



# Secondary ice production processes in wintertime alpine mixed-phase clouds

Paraskevi Georgakaki[1], Georgia Sotiropoulou[1,2], Étienne Vignon[3], Anne-Claire Billault-Roux[4], Alexis Berne[4] and Athanasios Nenes[1,5]

[1]Laboratory of Atmospheric Processes and their Impacts, School of Architecture, Civil & Environmental Engineering, École Polytechnique Fédérale de Lausanne, Lausanne, CH-1015, Switzerland
[2]Department of Meteorology, Stockholm University & Bolin Center for Climate Research, Stockholm, Sweden
[3]Laboratoire de Météorologie Dynamique/IPSL/Sorbonne Université/CNRS, UMR 8539, Paris, France
[4]Environmental Remote Sensing Laboratory, School of Architecture, Civil & Environmental Engineering, École Polytechnique Fédérale de Lausanne, Lausanne, CH-1015, Switzerland
[5]Center for Studies of Air Quality and Climate Change, Institute of Chemical Engineering Sciences, Foundation for Research and Technology Hellas, Patras, GR-26504, Greece

*Correspondence to*: athanasios.nenes@epfl.ch

**Abstract**

Observations of orographic mixed-phase clouds (MPCs) have long shown that measured ice crystal number concentrations (ICNCs) can exceed the concentration of ice nucleating particles by orders of magnitude. Additionally, model simulations of alpine clouds are frequently found to underestimate the amount of ice compared with observations. Surface-based blowing snow, hoar frost and secondary ice production processes have been suggested as potential causes, but their relative importance and persistence remains highly uncertain. Here we study ice production mechanisms in wintertime orographic MPCs observed during the Cloud and Aerosol Characterization Experiment (CLACE) 2014 campaign at the Jungfraujoch site in the Swiss Alps with the Weather Research and Forecasting model (WRF). Simulations suggest that droplet shattering is not a significant source of ice crystals at this specific location – but break-up upon collisions between ice particles is quite active, elevating the predicted ICNCs by up to 3 orders of magnitude, which is consistent with observations. The initiation of the ice-ice collisional break-up mechanism is primarily associated with the occurrence of seeder-feeder events from higher precipitating cloud layers. The enhanced aggregation of snowflakes is found to drive secondary ice formation in the simulated clouds, the role of which is strengthened when the large hydrometeors interact with the primary ice crystals formed in the feeder cloud. Including a constant source of cloud ice crystals from blowing snow, through the action of the break-up mechanism, can episodically enhance ICNCs. Increases in secondary ice fragment generation can be counterbalanced by enhanced orographic precipitation, which seems to prevent explosive multiplication and cloud dissipation. These findings highlight the importance of secondary ice and "seeding" mechanisms – primarily falling ice from above and to a lesser



degree blowing ice from the surface – which frequently enhance primary ice and determine the
phase state and properties of MPCs.

## 1. Introduction

Understanding orographic precipitation is one of the most critical aspects of weather
forecasting in mountainous regions (Roe, 2005; Rotunno and Houze, 2007; Chow et al., 2013).
Orographic clouds are often mixed-phase clouds (MPCs), containing simultaneously
supercooled liquid water droplets and ice crystals (Lloyd et al., 2015; Lohmann et al., 2016;
Henneberg et al., 2017). In mid- and high-latitude environments almost all precipitation
originates from the ice phase (Field and Heymsfield, 2015; Mülmenstädt et al., 2015),
highlighting the importance of correctly simulating the amount and distribution of both liquid
water and ice (i.e., the liquid-ice phase partitioning) in MPCs (Korolev et al., 2017).
Our understanding of MPCs remains incomplete owing to the numerous and highly
nonlinear cloud microphysical pathways driving their properties and evolution (Morrison et al.,
2012). Due to the lower equilibrium water vapor pressure over ice crystals than over liquid
water, MPCs tend to glaciate over time through the Wegener-Bergeron-Findeisen (WBF)
process, which is the rapid ice crystal growth at the expense of the surrounding evaporating
cloud droplets (Bergeron, 1935; Findeisen, 1938). Another process that can trigger cloud
glaciation and has been shown to enhance precipitation over mountains is the seeder-feeder
mechanism (e.g., Roe, 2005). This mechanism has been observed in several field studies (e.g.,
Reinking et al., 2000; Purdy et al., 2005; Mott et al., 2014; Ramelli et al., 2021) and refers to
ice crystals falling from a high-level seeder cloud into a lower-level cloud (external seeder-
feeder event) or a lower-lying part of the same cloud (in-cloud seeder-feeder event), where they
act as seeds for the glaciation of clouds. Satellite products covering the 11-year period between
April 2006 and October 2017 indicated that seeding events are widespread over Switzerland,
occurring with a frequency of 31% of the total observations (Proske et al., 2021). Despite these
two mechanisms that can readily destabilize an orographic cloud, a high frequency of MPCs
have been reported under high updraft velocity conditions prevailing over the complex
mountainous terrain (e.g., in the Swiss Alps), where supercooled liquid droplets are generated
faster than depleted by depositional ice growth and riming, leading to persistent mixed-phase
conditions (Korolev and Isaac, 2003; Lohmann et al., 2016).
At temperatures between 0 ˚C and –38 ˚C, where mixed-phase conditions can occur,
primary ice formation in clouds is catalyzed by the presence of insoluble aerosols that act as



ice nucleating particles (INPs) (e.g., Hoose and Möhler, 2012, Kanji et al., 2017). However,
in-situ observations of MPCs forming over mountain-top research stations or near mountain
slopes regularly reveal that there is a mismatch between the scarcity of primary INPs and the
measured ice crystal number concentrations (ICNCs) – the latter being several orders of
magnitude more abundant (Rogers and Vali, 1987; Geerts et al., 2015; Lloyd et al., 2015; Beck
et al., 2018; Lowenthal et al., 2019; Mignani et al., 2019). Model simulations of alpine MPCs
frequently fail to reproduce the elevated ICNCs dictated by observations (Farrington et al.,
2016; Henneberg et al., 2017; Dedekind et al., 2021). The fact that primary ice cannot explain
the observed ICNCs in orographic MPCs has often been attributed to the influence of surface
processes such as the lofting of snowflakes (i.e., blowing snow; Rogers and Vali, 1987; Geerts
et al., 2015), detachment of surface hoar frost (Lloyd et al., 2015), turbulence near the mountain
surface or convergence of ice particles due to orographic lifting (Beck et al., 2018) and riming
on snow-covered surfaces (Rogers and Vali, 1987).

Among these surface processes, the impact of blowing snow ice particles (BIPS) has been

studied thoroughly, either using observations collected in mountainous regions (e.g., Lloyd et
al., 2015; Beck et al., 2018; Lowenthal et al., 2019), and detailed surface snow models (e.g.,
Lehning et al., 2006; Vionnet et al., 2013, 2014) or through remote sensing techniques (e.g.,
Rogers and Vali, 1987; Vali et al., 2012; Geerts et al., 2015). BIPS are found to hover close to
the surface provided that the wind speed exceeds a threshold value, which varies between 4
and 13 ms$^{-1}$ (e.g., Déry and Yau, 1999; Mahesh et al., 2003), depending on the snowpack
properties and the prevailing atmospheric conditions. The transport of BIPS is commonly
separated into the saltation layer and the turbulent suspension layer. The saltation layer is a
shallow layer formed close to the ground, where the transported ice particles are found to follow
ballistic trajectories. Turbulent eddies or upward gusts can then diffuse the saltated ice particles
up to a height of several tens of meters above the surface, into the suspension layer (e.g., Vali
et al., 2012; Vionnet et al., 2014).

In-cloud secondary ice production (SIP) processes may also enhance ice production after

the initial primary ice nucleation events. Especially for orographic clouds, whose cloud top
temperatures are not cold enough to activate sufficient INPs, ice multiplication through SIP
might be particularly important. Over the past few decades, several SIP mechanisms have
emerged in literature, a detailed review of which is provided by Field et al. (2017) and Korolev
and Leisner (2020). We briefly review the three main SIP mechanisms below.

The rime-splintering, also known as the Hallett-Mossop (HM) process (Hallett and

Mossop, 1974), is argued to be the most efficient one in slightly supercooled clouds (i.e.,



temperatures warmer than –10 ˚C). The HM process refers to the ejection of small secondary
ice splinters after a supercooled droplet with a diameter larger than ~25 μm rimes onto a large
ice particle at temperatures between –8 and –3 ˚C (Choularton et al., 1980; Heymsfield and
Mossop, 1984). Although this is the only SIP mechanism widely implemented in current
microphysics schemes (e.g., Beheng, 1987; Phillips et al., 2001; Morrison et al., 2005), recent
modeling studies of slightly supercooled polar clouds, have shown that it cannot sufficiently
explain the enhanced ICNCs in remote environments (Young et al., 2019; Sotiropoulou et al.,
2020, 2021a). Moreover, aircraft measurements have reported high ICNCs when the conditions
required for HM initiation are not fulfilled (e.g., Korolev et al., 2020).

A second process that is found to contribute to ice multiplication over a wider

temperature range is the collisional breakup (BR), which involves the fracturing of delicate ice
particles due to collisions with other ice particles (Vardiman, 1978; Griggs and Choularton,
1986; Takahashi et al., 1995). Evidence for this process is provided from several field studies
in the Arctic (Rangno and Hobbs, 2001; Schwarzenboeck et al., 2009) or in the Alps (Mignani
et al., 2019; Ramelli et al., 2021) and from limited laboratory investigations (Vardiman 1978;
Takahashi et al. 1995). These two studies created a basis for various numerical formulations of
the BR mechanism (e.g., Phillips et al., 2017a; Sullivan et al., 2018a; Sotiropoulou et al., 2020).
Parameterizations of this mechanism are implemented in small-scale models (Fridlind et al.,
2007; Phillips et al., 2017a, b; Sotiropoulou et al., 2020, 2021b; Sullivan et al., 2018a; Yano
and Phillips, 2011; Yano et al., 2016), mesoscale models (Hoarau et al., 2018; Sullivan et al.,
2018b; Qu et al., 2020; Sotiropoulou et al., 2021a; Dedekind et al., 2021) and global climate
models (Zhao and Liu, 2021). These modeling studies followed several approaches to
implement the effect of BR. For instance, Hoarau et al. (2018) assumed a constant number of
fragments generated per collision in the Meso-NH model, while Sullivan et al. (2018b)
implemented a temperature-dependent relationship in the COSMO-ART mesoscale model
based on the results of Takahashi et al. (1995). This simplified formulation was further
modified to account for the hydrometeor size scaling, which improved the representation of
ICNCs in alpine clouds (Dedekind et al., 2021). Sotiropoulou et al. (2020) and (2021a)
reproduced the observed ICNCs in polar clouds, by applying the physically-based
parameterization developed by Phillips et al. (2017a, b). At slightly colder temperatures
(between –12.5 ºC and –7 ºC), however, BR was found to be generally weak over the Arctic
(Sotiropoulou et al., 2021b; Zhao et al., 2021).

Droplet shattering (DS) during freezing is a third process that is frequently suggested to

explain the unexpected ice enhancement in clouds. This mechanism occurs when a drizzle-



sized droplet, with a diameter larger than ~50 μm collides with an ice particle or INP, triggering
its freezing after a solid ice shell is formed around the droplet (e.g., Griggs and Choularton,
1983). As the freezing moves inward, the pressure starts to build and the freezing droplet reacts
by either breakup in two halves, cracking, bubble burst or jetting (e.g., Keinert et al., 2020).
These processes may be accompanied by the ejection of small ice fragments, the number of
which is yet poorly constrained as recent laboratory studies are showing a large diversity of
results (Lauber et al., 2018; Keinert et al., 2020; Kleinheins et al., 2021). Individual
experiments of freezing droplets reported the maximum fragmentation rate at temperatures
between ~ –10 and –15 °C (Leisner et al., 2014; Lauber et al., 2018; Keinert et al., 2020). DS
is found to be very efficient in vigorous convective updrafts (Lawson et al., 2015; Phillips et
al., 2018; Korolev et al., 2020; Qu et al., 2020), while remote sensing observations indicate that
DS can be much more conducive to SIP in slightly supercooled Arctic MPCs than the well-
known HM process (Luke et al., 2021). This is in line with single-column simulations
performed by Zhao et al. (2021), but contradicts the findings of small-scale modeling studies
suggesting that DS is ineffective in polar regions (Fu et al., 2019; Sotiropoulou et al., 2020).
Mesoscale model simulations of winter alpine clouds formed at temperatures lower than –8 °C
indicate that DS is not contributing to the modeled ICNCs (Dedekind et al., 2021), while field
observations suggest the increasing efficiency of the mechanism at temperatures warmer than
–3 °C (Lauber et al., 2021).

In the orographic MPCs observed during the Cloud and Aerosol Characterization
Experiment (CLACE) 2014 campaign at the high-altitude research station of Jungfraujoch
(JFJ) in the Swiss Alps, the measured ICNCs exceeded the predicted INPs by 3 orders of
magnitude, reaching up to ~1000 L$^{-1}$ at temperatures around –15 °C (Lloyd et al. 2015). Whilst
ice multiplication through BR and DS mechanisms show a peak production around a similar
temperature, Lloyd et al (2015) did not find evidence for their occurrence. Instead, they
suggested that at periods when there was a strong correlation between horizontal wind speed
and observed ICNCs, BIPS is contributing to the latter, but the mechanism was incapable of
producing ICNCs higher than ~100 L$^{-1}$. In the absence of such correlation, a flux of hoar frost
crystals was considered responsible for the very high ice concentration events (ICNCs > 100
L$^{-1}$), albeit without any direct evidence. Beck et al. (2018) argued that the relationship between
ICNCs and horizontal wind speed may not be a good indicator for distinguishing between
blowing snow and hoar frost. Their measurements conducted at the Sonnblick Observatory in
the Austrian Alps revealed the presence of several hundred ice crystals of blowing snow per
liter during cloud-free conditions. In a cloudy environment, though, such high contribution





from BIPS was found only close to the surface, with the concentrations dropping to several
tens to 100 L$^{-1}$ at heights above ~10 m.

From a modeling perspective, the causes of the surprisingly high ICNCs in orographic

MPCs formed during the CLACE 2014 campaign were explored in Farrington et al. (2016).
Since temperatures at JFJ are generally outside the HM temperature range ($< -8$ °C), Farrington
et al. (2016) used back trajectories analysis to investigate whether splinters produced at lower
altitudes through the HM process could be lifted to the summit of JFJ elevating the modeled
ICNCs. They showed that the inclusion of the HM process upwind of JFJ could not explain the
measured concentrations of ice, while the addition of a surface flux of hoar crystals provided
the best agreement with observations. Although surface-originated processes have been
frequently invoked to explain the disparity between ICNCs and INPs, the role of SIP processes
– especially the BR and the DS mechanism – has received much less attention. In this study
we utilize the Weather Research and Forecasting model (WRF) to conduct simulations of two
case studies observed in winter during the CLACE 2014 campaign. Our primary objective is
to investigate if the implementation of two SIP parameterizations that account for the effect of
BR and DS can reduce the discrepancies between observed and simulated ICNCs. Additionally,
we aim to identify the conditions favoring the initiation of SIP in the orographic terrain and
explore the synergistic influence of SIP with wind-blown ice.

**2. Methods**
2.1 CLACE instrumentation
CLACE is a long-established series of campaigns taking place for over two decades at the
mountain-top station of JFJ, located in the Bernese Alps, in Switzerland, at an altitude of ~3580
m above sea level (a.s.l.) (e.g., Choularton et al., 2008). The measurement area is very complex
and heterogeneous with distinct mountain peaks (Fig. 1), while JFJ is covered by clouds
approximately 40% of the time, offering an ideal location for microphysical observations
(Baltensperger et al., 1998). Owing to the local orography surrounding the site, the wind flow
is constrained to two directions (Ketterer et al., 2014). Under southeasterly (SE) wind
conditions, air masses are lifted along the moderate slope of the Aletsch Glacier, whereas under
northwesterly (NW) wind conditions the air is forced to rise faster along the steep north face
of the Alps, which is associated with persistent MPCs (Lohmann et al., 2016). A detailed
description of the in-situ and remote sensing measurements taken during January and February



2014 as part of the CLACE 2014 campaign is provided by Lloyd et al. (2015) and Grazioli et
al. (2015). Here we only offer a brief presentation of the datasets used in this study.

Shadowgraphs of cloud particles were produced by the two-dimensional stereo

hydrometeor spectrometer (2D-S; Lawson et al., 2006), part of a three-view cloud particle
imager (3V-CPI) instrument. The 2D-S products have been used to provide information on the
number concentration and size distribution of particles in the size range of 10-1280 μm.
Following Crosier et al. (2011), the raw data were further processed to determine between ice
crystals and droplets, and to remove artefacts from shattering events (Korolev et al., 2011). An
approximation of the ice water content (IWC) at JFJ could also be derived by the 2D-S data
using the Brown and Francis (1995) mass-diameter relationship with a factor of up to 5
uncertainty (Heymsfield et al., 2010). Additionally, the quantification of the liquid water
content (LWC) is based on the liquid droplet size distribution data derived from a DMT cloud
droplet probe (CDP; Lance et al., 2010) over the size range between 2 and 50 μm. Typical
meteorological parameters (e.g., temperature, relative humidity, wind speed and wind
direction), that served as comparison to assess the validity of the model, were provided by the
weather station managed by MeteoSwiss at JFJ. The instrumentation was set up on the roof
terrace outside the Sphinx laboratory.

2.2 WRF simulations
WRF model, version 4.0.1, with augmented cloud microphysics (Sotiropoulou et al., 2021a) is
used for non-hydrostatic cloud-resolving simulations. The model has been run with three two-
way nested domains (Fig. 1), with a respective horizontal resolution of 12, 3 and 1 km. Two-
way grid nesting is generally found to improve the model performance in the inner domain
(e.g., Harris and Durran, 2010), although the sensitivity of the results to the applied nesting
technique has been shown to be negligible (not shown). The parent domain consists of 148×148
grid points centered over the JFJ station (46.55°N, 7.98°E, shown with a black dot in Fig. 1),
while the second and the third domain include 241×241 and 304×304 grids, respectively. The
Lambert conformal projection is applied to all three domains, as it is well-suited for mid-
latitudes. Here we adapted the so-called "refined" vertical grid spacing proposed by Vignon et
al. (2021), using 100 vertical eta levels up to a model top of 50 hPa (i.e., ~20 km). This set-up
provides a refined vertical resolution of ~100 m up to mid-troposphere at the expense of the
coarsely resolved stratosphere. To investigate the dynamical influence on the development of
MPCs under the two distinct wind regimes prevailing at JFJ (Section 2.1), we simulate two



case studies, starting on 25 January and 29 January 2014, 00:00 UTC, respectively. Both case
studies are associated with the passage of frontal systems over the region of interest,
approaching the alpine slopes either from the NW (cold front) or the SE (warm front) direction,
as shown by the vertically-integrated condensed water content (ICWC; sum of cloud droplets,
rain, cloud ice, snow, and graupel) in Fig. 1. For both cases the simulation covers a 3-day
period, with the first 24 hours being considered sufficient time for spin up. A 27-s time step
was used in the parent domain and goes down to 9 s in the second domain and 3 s in the third
domain. Note that achieving such small time steps in the innermost domain is essential to
ensure numerical stability in non-hydrostatic simulations over a region with complex
orography such as around JFJ.

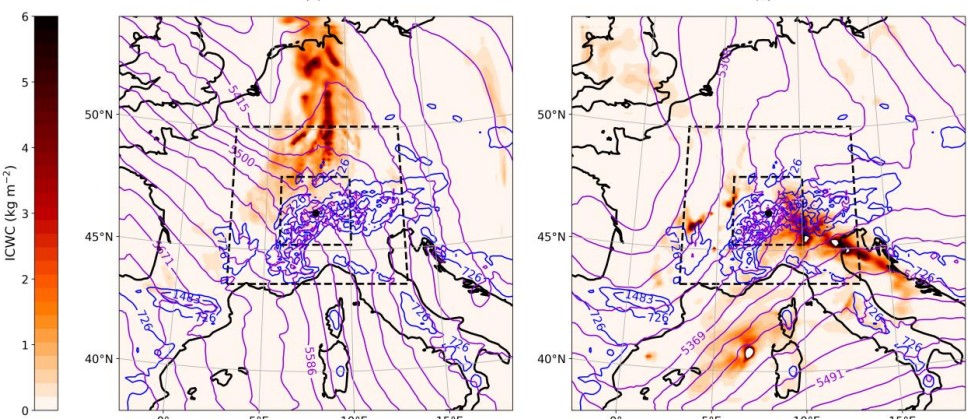

**Figure 1.** Map of synoptic conditions around JFJ station at (a) 00:00 UTC, 26 January 2014
and (b) 00:00 UTC, 30 January 2014, from the control simulation (12-km resolution domain).
The purple (blue) contours show the 500 hPa geopotential height in m (the terrain heights in
m). The color shading shows the vertically-integrated condensed water content (in kg m$^{-2}$). The
black dashed lines delimit the 3-km and 1-km resolution domains, while the black dot locates
the JFJ station.

The ERA5 reanalysis dataset (Hersbach et al., 2020) is used to initialize the model and
provide the lateral forcing at the edge of the 12-km resolution domain every 6 hours. Static
fields at each model grid point come from default WRF pre-processing system datasets, with a
resolution of 30″ for both the topography and the 'land use' fields. The MODIS-based dataset
is used for land cover. Regarding the physics options chosen to run WRF simulations, the Rapid
Radiative Transfer Model for general circulation models (RRTMG) radiation scheme is applied
to parameterize both the short-wave and long-wave radiative transfer. The vertical turbulent



mixing is treated with the Mellor-Yamada-Janjic (MYJ; Janjić, 2002) 1.5 order scheme, while
surface options are modeled by the Noah land-surface model (Noah LSM; Chen and Dudhia,
2001). The Kain-Fritsch cumulus parameterization has been activated only in the outermost
domain, as the resolution of the two nested domains is sufficient to reasonably resolve cumulus-
type clouds at grid-scale.

*2.2.1 Microphysics scheme and primary ice production*
The Morrison two-moment scheme (Morrison et al., 2005; hereafter M05) is used to
parameterize the cloud microphysics, following the alpine cloud study of Farrington et al.
(2016). The scheme includes double-moment representations of rain, cloud ice, snow and
graupel species, while cloud droplets are treated with a single-moment approach and therefore
the droplet number concentration ($N_d$) must be prescribed. Here $N_d$ is set to 100 cm$^{-3}$, based on
the mean $N_d$ observed within the simulated temperature range (Lloyd et al., 2015).

Three primary ice production mechanisms through heterogeneous nucleation are

described in the default version of the M05 scheme, namely immersion freezing, contact
freezing, and deposition/condensation freezing nucleation. Immersion freezing of cloud
droplets and raindrops is described by the probabilistic approach of Bigg (1953). Contact
freezing is parameterized following Meyers et al. (1992). Finally, deposition and condensation
freezing is represented by the temperature-dependent equation derived by Rasmussen et al.
(2002) based on the in-situ measurements of Cooper (1986) collected from different locations
at different temperatures. Following Thompson et al. (2004), this parameterization is activated
either when there is saturation with respect to liquid water and the simulated temperatures are
below –8 °C or when the saturation ratio with respect to ice exceeds a value of 1.08. The
accuracy of these parameterizations in representing atmospheric INPs is debatable as they are
derived from very localized measurements over a limited temperature range. Nevertheless,
Farrington et al. (2016) argued that the deposition/ condensation freezing parameterization of
Cooper (1986) can effectively explain INPs between the range 0.01 and 10 L$^{-1}$, which is
frequently observed during field campaigns at JFJ (Chou et al., 2011; Conen et al., 2015).

*2.2.2 Ice multiplication through rime splintering in the M05 scheme*
Apart from primary ice production, the HM process is the only SIP mechanism included in the
default version of the M05 scheme. This parameterization adapted from Reisner et al. (1998),
based on the laboratory findings of Hallett and Mossop (1974), allows for  splinter production
after cloud- or rain- drops are collected by rimed snow particles or graupels. The efficiency of
this process is zero outside the temperature range between –8 and –3 °C, while it follows a
linear temperature-dependent relationship in between. HM is not activated unless the rimed ice
particles have masses larger than 0.1 g kg$^{-1}$ and cloud or rain mass mixing ratio exceeds the
value of 0.5 g kg$^{-1}$ or 0.1 g kg$^{-1}$, respectively. Since these conditions are rarely met in natural
MPCs, previous modeling studies had to artificially remove any thresholds to achieve an
enhanced efficiency of this process (Young et al., 2019; Atlas et al., 2020). In the current study,
however, the HM process is not effective, as the simulated temperatures at JFJ altitude are
below –8 °C (see Sect. 2.3).

*2.2.3 Ice multiplication through ice-ice collisions in the M05 scheme*
In addition to the HM process, we have also included two parameterizations to represent the
BR mechanism. An extensive description of the implementation method is provided in
Sotiropoulou et al. (2021a) (see their Appendix B). Among the three ice particle types included
in the M05 scheme (i.e., cloud ice, snow, graupel), we assume that only the collisions between
cloud ice-snow, cloud ice-graupel, graupel-snow, snow-snow, and graupel-graupel can result
in ice multiplication. The first parameterization tested here follows the simplified methodology
proposed by Sullivan et al. (2018a), which is based on the laboratory work of Takahashi et al.
(1995). Their findings revealed a strong temperature dependence of the fragment numbers
generated per collision ($N_{BR}$):

$$N_{BR} = 280\,(T - T_{min})^{1.2} e^{-(T-T_{min})/5}\,, \tag{1}$$

where $T_{min} = 252\,K$, is the minimum temperature for which BR occurs. Yet their experimental
set-up was rather simplified involving only collisions between large hail-sized ice spheres with
diameters of ~2 cm. Taking this into account, Sotiropoulou et al. (2021a) further scaled the
temperature-dependent formulation for size:

$$N_{BR} = 280\,(T - T_{min})^{1.2} e^{-(T-T_{min})/5}\,\frac{D}{D_0}\,, \tag{2}$$

where $D$ is the size in meters of the particle that undergoes break-up and $D_0$=0.02 m is the size
of the hail-sized balls used in the experiments of Takahashi et al. (1995).

Phillips et al. (2017a) proposed a more physically-based formulation, developing an

energy-based interpretation of the experimental results conducted by Vardiman (1978) and
Takahashi et al. (1995). The initial collisional kinetic energy is considered as the governing
constraint driving the BR process. Moreover, the predicted $N_{BR}$ depends on the ice particle type



and morphological habit and is a function of the temperature, particle size and rimed fraction.
Here the generated fragments per collision are described as follows:

$$N_{BR} = aA \left(1 - exp\left\{-\left[\frac{CK_0}{aA}\right]^\gamma\right\}\right), \tag{3}$$

where $K_0 = \frac{1}{2}\frac{m_1 m_2}{m_1+m_2} (\Delta u_{n12})^2$ is the initial kinetic energy, in which $m_1$ and $m_2$ are the masses
of the colliding particles and $|\Delta u_{n12}| = \{(1.7\, u_{n1} - u_{n2})^2 + 0.3\, u_{n1}\, u_{n2}\}^{1/2}$ is the difference
in their terminal velocities. The correction term is proposed by Mizuno et al. (1990) and Reisner
et al. (1998) to account for underestimates when $u_{n1} \approx u_{n2}$. The parameter $a$ in Eq. 3 is the
surface area of the smaller ice particle (or the one with the lower density), defined as $a = \pi D^2$,
with $D$ as in Eq. 2. $A$ in Eq. 3 represents the number density of breakable asperities on the
colliding surfaces. For collisions that involve cloud ice and snow particles $A$ is described as
$A = 1.58 \times 10^7 (1 + 100\Psi^2)(1 + \frac{1.33\times 10^{-4}}{D^{1.5}})$, where $\Psi < 0.5$ is the rimed fraction of the most
fragile ice particle. For graupel-graupel collisions $A$ is given by a temperature-dependent
equation as $A = \frac{a_0}{3} + \max(\frac{2a_0}{3} - \frac{a_0}{9}|T - 258|, 0)$, in which $a_0 = 3.78 \times 10^4 \left(1 + \frac{0.0079}{D^{1.5}}\right)$. $C$
is the asperity-fragility coefficient, which is empirically derived to account for different
collision types, while the exponent $\gamma$ is equal to 0.3 for collisions between graupel-graupel and
is calculated as a function of the rimed fraction for collisions including cloud ice and snow.
The parameterization was developed based on particles with diameters 500 μm < $D$ < 5 mm,
however Phillips et al. (2017a) suggest that it can be used for particle sizes outside the
recommended range as long as the input variables to the scheme are set to the nearest limit of
the range. Finally, since $N_{BR}$ was never observed to exceed 100 in the experiments of Vardiman
(1978), here we also use this value as an upper limit for all collision types (Phillips et al.,
2017a). All predicted fragments emitted through BR are added to the cloud ice category.

*2.2.4 Ice multiplication through droplet shattering in the M05 scheme*
Two different parameterizations are implemented in the M05 scheme to investigate the
potential efficiency of the DS mechanism in producing secondary ice splinters ($N_{DS}$). Phillips
et al. (2018) proposed two possible modes of raindrop-ice collisions, that can initiate the
freezing process. In the first mode, the freezing of the drop occurs either by collecting a small
ice particle or through heterogeneous freezing. In the default M05 scheme, the product of the
collisions between raindrops and cloud ice is considered to be graupel (snow) – if the rain
mixing ratio is greater (lower) than 0.1 g kg$^{-1}$, following Reisner et al. (1998). Additionally,





the heterogeneous freezing of big raindrops in the immersion mode follows Bigg's (1953)
parameterization (Section 2.2.1). Here we consider that the product of these two processes can
undergo shattering and generate numerous ice fragments, the number of which is parameterized
after Phillips et al. (2018). The formulation is derived by fitting to a pooled dataset from
laboratory studies and is given as a Lorentzian function of temperature and a polynomial
expression of the drop size. More precisely, in the first mode of the formulation, the total
number of fragments ($N$) generated per frozen drop are given by:

$$N = \Xi(D_r)\Omega(T)\left[\frac{\zeta\eta^2}{(T-T_0)^2+\eta^2} + \beta T\right],\tag{4}$$

where T is the temperature (in K) and $D_r$ is the size of the freezing raindrop (in mm). Note that
$N$ is defined as the sum of the big fragments ($N_B$) and tiny splinters ($N_T$). Equation (4) applies
only to drop diameters less than 1.6 mm, which is the maximum observed experimentally. For
droplet sizes beyond this maximum value, $N$ can be inferred by linear extrapolation. $N_B$ is
described by another Lorentzian:

$$N_B = min\left\{\Xi(D_r)\Omega(T)\left[\frac{\zeta_B\eta_B{}^2}{\left(T-T_{B,0}\right)^2+\eta_B{}^2}\right], N\right\}.\tag{5}$$

The factors $\Xi(D_r)$ and $\Omega(T)$ in Eq. (4) and (5) are cubic interpolation functions, preventing the
onset of DS for $D_r < 0.05$ mm and T > –3 °C. Furthermore, the parameters $\zeta$, $\eta$, $T_0$, $\beta$, $\zeta_B$, $\eta_B$,
$T_{B,0}$, found in these relationships, are derived from previous laboratory studies and are
analytically described in Phillips et al. (2018). Note that the big fragments emitted (i.e., $N_B$)
will be initiated in the model as graupel, snow or frozen drops, while it is only the tiny splinters
($N_T = N - N_B$) that are considered secondary ice (i.e., $N_{DS} = N_T$) and will be passed to the cloud
ice category.

The second mode of raindrop-ice collisions includes the accretion of raindrops on impact

with more massive ice particles, such as snow or graupel, the description of which in the M05
scheme is adapted from Ikawa and Saito (1991). This mode has been studied only once in the
laboratory study of Latham and Warwicker (1980), who reported that the collision of
supercooled raindrops with hailstones can potentially stimulate secondary ice. Since there was
no quantitative observation of this mode, Phillips et al. (2018) proposed an empirical, energy-
based formulation to account for the tiny splinters ejected after collisions between raindrops
and large ice particles:

$$N_{DS} = 3\Phi(T) \times [1 - f(T)] \times max(DE - DE_{crit}, 0)\ ,\tag{6}$$



where $DE = \frac{K_0}{S_e}$, is the dimensionless energy given as the ratio of the initial kinetic energy ($K_0$;
described in 2.2.3) over the surface energy, which is expressed by the product $S_e = \gamma_{liq}\pi D_r{}^2$,
in which $\gamma_{liq}$=0.073 J m$^{-2}$, is the surface tension of liquid water. The critical value of DE used
in Eq. (6) for the onset of splashing upon impact is set to $DE_{crit} = 0.2$. The parameter $f(T) =$
$-c_w T/L_f$, represents the initial frozen fraction of a supercooled drop during the first stage of
the freezing process, where $C_w = 4200$ J kg$^{-1}$ K$^{-1}$, is the specific heat capacity of liquid water,
$L_f = 3.3 \times 10^5$ J kg$^{-1}$, is the specific latent heat of freezing, while T is the initial freezing
temperature ($^o$C) of the raindrop. Finally, $\Phi(T) = \min[4f(T),1]$ is an empirical fraction, which
represents the probability of any new drop in the splash products to contain a frost secondary
ice particle. At temperatures ~ –10 $^o$C this formulation yields $\Phi = 0.5$, meaning that the
probability of a secondary drop to contain ice is 50%. The first laboratory investigation of this
rather uncertain parameter as a function of temperature is provided in James et al. (2021).
Further details regarding the derivation of the empirical parameters and the uncertainties
underlying the mathematical formulations are discussed in Phillips et al. (2018).

Following Sullivan et al. (2018a), the second DS parameterization tested in this study is

described as the product of a polynomial expression of the freezing droplet size (Lawson et al.,
2015), a shattering probability ($p_{sh}$) and a freezing probability ($p_{fr}$):

$$N_{DS} = 2.5 \times 10^{-11} (D_r)^4 \, p_{sh} \, p_{fr} \ . \tag{7}$$

The $p_{sh}$ is based upon droplet levitation experiments shown in Leisner et al. (2014) and is
represented by a temperature-dependent Gaussian distribution, centered at ~ –15 $^o$C. Note that
$p_{sh}$ is non-zero only for droplets with sizes greater than 50 μm. The $p_{fr}$ is 0 for temperatures
warmer than –3 $^o$C and 1 if temperatures fall below –6 $^o$C, following the cubic interpolation
function, $\Omega(T)$, adapted from Phillips et al. (2018).

2.3 Model validation
The control simulation (CNTRL), performed with the standard M05 scheme, sets the basis for
assessing the validity of the model against available meteorological observations. Temperature,
relative humidity, wind speed, and wind direction are obtained from the MeteoSwiss weather
station at JFJ. The comparison of each meteorological variable with the results from the nearest
model grid point of the CNTRL simulation is shown in Fig. 2. Note that the outputs are from
the first atmospheric level of the innermost domain at ~10 m above ground level (a.g.l) (Fig.
1), while the first 24 hours of each simulation period are considered spin up time and are





therefore excluded from the present analysis. The mean modeled values and standard
deviations (std), along with the root mean square error (RMSE) and the index of agreement
(IoA) between model predictions and observational data are summarized in Table 1. IoA is
both a relative and a bounded measure (i.e., $0 \leq IoA \leq 1$) that describes phase errors between
predicted ($P_i$) and observed ($O_i$) time series (Willmott et al., 2012):

$$IoA = 1 - \left[ \frac{\sum_{i=1}^{N}(P_i - O_i)^2}{\sum_{i=1}^{N}(|P_i'| - |O_i'|)^2} \right] ,\tag{8}$$

where $P_i' = P_i - \bar{O}$ and $O_i' = O_i - \bar{O}$, in which $\bar{O}$ is the mean of the observed variable.

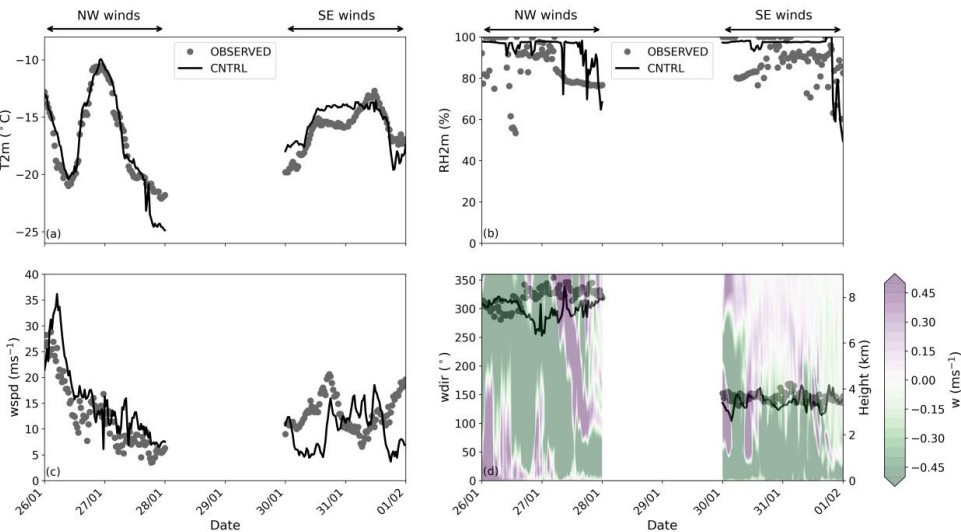

**Figure 2.** Time series of (a) temperature (T2m), (b) relative humidity with respect to liquid
phase at 2m height (RH2m), (c) wind speed (wspd) and (d) wind direction (wdir). Grey circles
indicate measurements collected between 26 January and 1 February 2014 at JFJ station, while
modeled values from CNTRL simulation are shown with a black line. The semi-transparent
contour plot is representing the vertical velocity (w) profile predicted by the CNTRL
simulation. Each day starts at 00:00 UTC.

Throughout the two case studies, the WRF simulations seem to closely follow the
observed temperatures (Fig. 2a), which is also indicated by the high IoA in Table 1. The
synoptic situation occurring on 26 January, with a deep trough extending to western Europe
(Fig. 1), has been associated with intense snowfalls in the alpine regions (Panziera and Hoskins,
2008). The passage of the cold front was followed by a sharp temperature decrease, with the
simulated temperatures fluctuating between –10 and ~ –20 °C throughout the NW case (Fig.
2a). Under the influence of the warm front during the SE case, the modeled temperatures rose





from ~ –18 ºC to ~ –14 ºC and remained less variable until 30 January 12:00 UTC, with mean
values of ~ –15.5 ºC (Table 1).

**Table 1.** Mean modeled values (± standard deviations), RMSE and IoA between the CNTRL
simulation of WRF and measurements carried out by the MeteoSwiss station at JFJ.

| Variable | Mean ± std | | RMSE | | IoA | |
|---|---|---|---|---|---|---|
| | NW winds | SE winds | NW winds | SE winds | NW winds | SE winds |
| T2m (˚C) | -17.10 ± 4.36 | -15.48 ± 1.75 | 1.40 | 1.33 | 0.97 | 0.84 |
| RH2m (%) | 94.07 ± 7.02 | 94.24 ± 10.31 | 14.01 | 11.61 | 0.55 | 0.64 |
| wspd (ms$^{-1}$) | 15.57 ± 7.45 | 9.78 ± 3.94 | 4.85 | 6.75 | 0.88 | 0.22 |


Fig. 2c and 2d reveal that the 1-km resolution domain can sufficiently capture the local
wind systems to a certain extent. During the NW flow, the horizontal wind speeds are
reproduced better by the CNTRL simulation (IoA=88%), whereas during the SE winds, the
simulated wind speed is frequently underestimated compared with observations (Fig. 2c). Such
deviations in the horizontal wind speed might be caused by the relatively coarse horizontal
resolution of the model, which prevents some small-scale and very local orographic structures
from being resolved. As discussed in Section 2.2, the observed winds at JFJ are channeled by
the orography to either NW or SE directions. The CNTRL simulation of WRF can satisfactorily
reproduce the wind direction in both cases, although the simulated values exhibit larger
fluctuations than the measured ones (Fig. 2d), presumably because of the surrounding
orography being less accurately represented in the model. This is particularly evident during
NW winds, when the simulated wind directions shift slightly to west directions compared to
observations. The positive vertical velocities, illustrated in the contour plot in Fig. 2d, result
from the orographically forced lifting of the airmasses over the local topography, and are not
related to convective instability in the lower atmospheric levels. The stronger updrafts
prevailing until the end of 26 January are associated with the steep ascent of the air parcels,
which can also contribute to the enhanced relative humidity (Fig. 2b). After the frontal passage,
the vertical velocities at the lower levels are downward directed, with the vertical profile of
potential temperature revealing that the atmosphere at JFJ is stabilized (not shown). The same
vertical velocity pattern, with mainly downward motions, characterizes the stably stratified





atmosphere after 30 January. Overall, Fig. 2 suggests that local meteorological conditions at
JFJ are reasonably well represented by the model.

2.4 Model simulations
Given the good representation of the atmospheric conditions at JFJ, the CNTRL simulation of
WRF is further accompanied by four sensitivity simulations, aiming to investigate the
contribution of BR and DS mechanisms. Here we also perform three additional sensitivity
experiments to explore the potential impact of blowing ice and the synergistic interaction with
SIP on the development of the simulated MPCs. A detailed list of the sensitivity experiments
is provided in Table 2.

The contribution of the DS mechanism is addressed in two sensitivity experiments,

DS_PHILL and DS_SULL, where the parameterizations of Phillips et al. (2018) and Sullivan
et al. (2018a) were applied, respectively (Section 2.2.4). Both sensitivity simulations yield
predictions that coincide with the CNTRL simulation (supplement Fig. S1) suggesting that the
DS mechanism is hardly ever activated, and fail to produce realistic total ice number
concentrations ($N_{isg}$; cloud ice + snow + graupel). The absence of correlation between LWC
and $N_{isg}$ fluctuations might also suggest the ineffectiveness of this mechanism under the
examined conditions. Note that the parameterized expressions used to describe the DS
mechanism involve a number of empirical and rather uncertain parameters, the value of which
could potentially influence the efficiency of the process in producing secondary ice fragments.
However, the sensitivity of our results to the choice of these parameters would be negligible,
as the low concentrations ($\lesssim 10\text{-}2$ cm-3) of relatively small raindrops with mode diameters
below the threshold size of 50 μm seem to completely prevent the onset of the DS process
(supplement Fig. S2). The DS mechanism is therefore excluded from the following discussion.
This result is in line with the modeling study of Dedekind et al. (2021), who also highlighted
the inefficiency of this mechanism in wintertime alpine clouds.

Two additional sensitivity simulations are conducted to investigate if the BR mechanism

can account for the observed ICNCs. First, the temperature-dependent formula of Takahashi et
al. (1995) scaled with the size of the particles that undergo fragmentation (Sotiropoulou et al.
2021a) is tested in the TAKAH simulation. The PHILL simulation is then conducted to test the
performance of the more advanced Phillips et al. (2017a) parameterization. Note that the
parameters involved in the parameterized expression of $N_{BR}$ (Eq. 3) concern the effect of ice
habit and rimed fraction of the colliding ice particles – which is not explicitly resolved in the

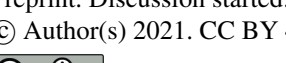


M05 scheme. Regarding the ice habit, we assume spatial planar ice particles, based on the
imagery presented in Lloyd et al. (2015) and Grazioli et al. (2015), which revealed the
predominance of sectored plates and oblate particles (probably columns or needles), along with
some rimed hydrometeors, at temperatures ~ –15 ºC. Balloon-borne measurements taken in
low-level orographic MPCs within seeder-feeder events revealed the presence of a large
fraction of graupel and rimed particles (Ramelli et al., 2021). For this reason, during the NW
and SE cases we consider rimed fractions of 0.4 and 0.3 to account for heavily and moderately
rimed ice particles, respectively. A higher rimed fraction is prescribed for the NW-winds case
study though, as the co-existence of ice crystals and liquid droplets under the stronger updraft
conditions are expected to favor ice crystal growth through riming. However, the sensitivity of
our results to the rimed fraction was not found significant.

**Table 2.** List of sensitivity simulations conducted with the WRF model.

| Simulation | BR process | DS process | NBIPS (L$^{-1}$) |
|---|---|---|---|
| CNTRL | off | off | 0 |
| DS_PHILL | off | Phillips et al., 2018 | 0 |
| DS_SULL | off | Sullivan et al., 2018a | 0 |
| TAKAH | Takahashi et al., 1995 | off | 0 |
| PHILL | Phillips et al., 2017a | off | 0 |
| BIPS10 | off | off | 10 |
| BIPS100 | off | off | 100 |
| BIPS100_PHILL | Phillips et al., 2017a | off | 100 |


The remaining sensitivity simulations focus on the potential impact of BIPS. Given that
in the default M05 scheme there is no parameterization of a flux of ice particles from the
surface, we parameterize the effect of BIPS lofting into the simulated orographic clouds by
applying a constant ice crystal source to the first atmospheric level of WRF over the whole
model domain. Although the source of BIPS at the first model level remained constant, yet
their number will be affected by processes such as advection, sublimation and sedimentation,
that are described in the M05 scheme. Note that the relatively coarse horizontal resolution in
the innermost domain of our simulations (i.e., 1 km) does not allow the accurate representation
of the small-scale turbulent flow over the orographic terrain. This is considered a limitation of
our methodology, since turbulent diffusion is a key process affecting the amount of BIPS that
will be resuspended from the surface.





The applied concentrations of BIPS varied between $10^{-2}$ and 100 $L^{-1}$, which is the upper
limit proposed by Lloyd et al. (2015) and observed within in-cloud conditions by Beck et al.
(2018). Number concentrations of BIPS (i.e., NBIPS) lower than 10 $L^{-1}$ were found incapable
of affecting the simulated cloud properties and are, therefore, not included in the following
discussion. Two sensitivity simulations are finally performed, BIPS10 and BIPS100 (Table 2),
in which the number indicates the NBIPS in $L^{-1}$. In our approach we assume BIPS are spherical
with diameters of 100 μm, based on typical sizes that are frequently reported in the literature
(e.g., Schlenczek et al., 2014; Schmidt, 1984; Geerts et al., 2015). The relatively small fall
speed of these particles (e.g., Pruppacher and Klett, 1997) will allow them to remain suspended
in the atmosphere. As a sensitivity we also considered smaller particles with sizes of 10 μm,
but our results did not change significantly (supplement Fig. S3). Besides, such small ice
particles are not expected to substantially contribute to the simulated IWC, as shown by
Farrington et al. (2016).
As SIP through BR and blowing snow are both important when trying to explain the high
ICNCs observed in alpine environments, their combined effect is addressed in our last
simulation, BIPS100_PHILL (Table 2). In this sensitivity simulation the effect of BR is
parameterized after Phillips et al. (2017a), while a constant ice crystal concentration of 100 $L^{-1}$
is applied to the first atmospheric level of WRF to represent the effect of BIPS.

**3.    Results and discussion**

3.1 Impact of SIP through BR on simulated microphysical properties
Fig. 3 displays the temporal evolution of the $N_{isg}$, IWC and LWC, at the first model level (~ 10
m a.g.l.) from the nearest to JFJ model grid point of the CNTRL, TAKAH and PHILL
simulations. Note that instead of focusing on a single grid point, we averaged the results from
the 9-km² area surrounding the point of interest. However, the produced time series showed
only little difference when compared to the nearest grid point time series, further validating the
robustness of our results (not shown). Besides, the region in the vicinity of JFJ is very
heterogeneous supporting the single point comparison presented in the following discussion.
The grey dots shown in Fig. 3 represent the measurements taken by the 2D-S and CDP
instruments at JFJ throughout the two periods of interest. The displayed time frequency of the
observations is 30 min to match the output frequency of the model. Note that the simulated
LWC includes liquid water from cloud droplets and rain, while the simulated IWC includes





cloud ice, snow and graupel. The contribution of rain in our simulations is, however, negligible
(supplement Fig. S2). Several statistical metrics for $N_{isg}$, IWC and LWC are summarized in
Table 3, 4 and 5, respectively. Note that periods with missing data in the measurement time
series are excluded from the statistical analysis.
During the NW flow, between 26 and 28 January, the measured ICNCs exceed 100 L$^{-1}$
for >50 % of the time, whereas during the SE flow the ICNCs usually fluctuate between 10 and
100 L$^{-1}$ (Fig. 3a). The highest ICNCs are generally observed at temperatures higher than ~ –15
℃, where SIP processes are thought to be dominant and primary ice nucleation in the absence
of bioaerosols is limited (e.g., Hoose and Möhler, 2012; Kanji et al., 2017). The CNTRL
simulation fails to reproduce $N_{isg}$ higher than 10 L$^{-1}$, with the mean simulated values being ~2-
2.5 L$^{-1}$ during both periods. At the same time the mean observed ICNC values are ~200 (70)
L$^{-1}$ during the NW (SE) case. Thus CNTRL systematically underestimates the amount of ice
by up to 2 orders of magnitude, which is also consistent with the interquartile statistics
presented in Table 3. With the HM process being totally ineffective in the prevailing
temperatures, this discrepancy suggests that ice crystals produced by heterogeneous ice
nucleation in CNTRL are not high enough to match the observations. A similar discrepancy
between predicted INPs and measured ICNCs was also documented in Lloyd et al. (2015).

**Table 3.** The 25$^{th}$, 50$^{th}$ (median) and 75$^{th}$ percentiles of ICNC time series (in L$^{-1}$).

| Simulation | 25$^{th}$ perc. | | Median | | 75$^{th}$ perc. | |
|---|---|---|---|---|---|---|
| | NW winds | SE winds | NW winds | SE winds | NW winds | SE winds |
| OBSERVED | 8.69 | 6.64 | 80.47 | 34.53 | 261.25 | 88.69 |
| CNTRL | 0.76 | 0.84 | 1.68 | 2.02 | 2.80 | 3.60 |
| TAKAH | 2.27 | 1.08 | 9.85 | 122.56 | 362.51 | 358.38 |
| PHILL | 2.49 | 0.76 | 6.27 | 2.09 | 118.21 | 59.23 |
| BIPS10 | 1.60 | 1.90 | 2.42 | 2.72 | 3.30 | 4.78 |
| BIPS100 | 6.17 | 10.74 | 10.36 | 13.88 | 12.32 | 17.39 |
| BIPS100_PHILL | 8.95 | 11.51 | 15.87 | 16.30 | 138.92 | 98.43 |


Activating the BR process in TAKAH and PHILL simulations is found to produce $N_{isg}$
as high as 400 L$^{-1}$ during both case studies (Fig. 3a), resulting in a substantially better
agreement with observations. At times when the simulated temperatures drop below ~ –18 ℃,
the $N_{isg}$ modeled by both simulations coincide with the CNTRL simulation. At relatively
warmer subzero temperatures though, the significant contribution of the BR process is evident,



elevating the predicted $N_{isg}$ by up to 3 orders of magnitude during the NW case and by more
than 2 orders of magnitude during the SE case. Although the median $N_{isg}$ in TAKAH and
PHILL remains underestimated compared to observations during the NW flow, the first seems
to produce unrealistically high median and 75[th] percentile values during the SE flow (Table 3).
Indeed, focusing on the $N_{isg}$ time series (Fig. 3a) TAKAH is ~25% of the time shown to
overestimate the observed ICNCs by a factor of ~3, reaching up a factor of 10 on 30 January
at 00:00. PHILL, on the other hand, produces more reasonable $N_{isg}$ throughout both case
studies, increasing $N_{isg}$ in the 75[th] percentile by more than 100 (50) L$^{-1}$ during the NW (SE)
case study (Table 3), reducing the gap between observations and model predictions.

**Table 4.** The 25[th], 50[th] (median) and 75[th] percentiles of IWC (in gm$^{-3}$) time series.

| Simulation | 25[th] perc. | | Median | | 75[th] perc. | |
|---|---|---|---|---|---|---|
| | NW winds | SE winds | NW winds | SE winds | NW winds | SE winds |
| OBSERVED | $31.5\times10^{-3}$ | $22.0\times10^{-3}$ | 0.63 | 0.24 | 0.66 | 0.26 |
| CNTRL | $4.3\times10^{-3}$ | $5.0\times10^{-3}$ | 0.03 | 0.04 | 0.15 | 0.12 |
| TAKAH | $1.3\times10^{-3}$ | $2.0\times10^{-3}$ | 0.10 | 0.09 | 0.52 | 0.34 |
| PHILL | $3.8\times10^{-3}$ | $3.7\times10^{-3}$ | 0.10 | 0.02 | 0.38 | 0.30 |
| BIPS100_PHILL | $3.9\times10^{-3}$ | $9.0\times10^{-3}$ | 0.09 | 0.03 | 0.40 | 0.31 |


The observed IWC time series (Fig. 3b) are frequently reaching ~1 gm$^{-3}$ during the NW
case, with the median values being a factor of 2.5 higher than those observed during the SE
case (Table 4). This highlights the presence of more massive ice particles when higher updraft
velocities prevail. The CNTRL simulation cannot produce IWC values > 0.8 gm$^{-3}$ and is most
of the time below the observed range. Adding a description of the BR process in TAKAH and
PHILL simulations sufficiently increases the modeled IWC by up to ~1 order of magnitude
between 26 January 12:00 UTC and 27 January 06:00 UTC, when the modeled $N_{isg}$ exceeds
100 L$^{-1}$ and the temperatures remain higher than –16 °C. The same conditions are observed in
the SE case, between 12:00 and 18:00 UTC on 30 January, when IWC shows a ~3 fold
enhancement reaching the observed levels. The IWC values in the third quartile predicted by
TAKAH and PHILL are more than a factor of 2 higher than the ones predicted by CNTRL
(Table 4). This increase improves the model performance although the modeled IWC remains
slightly underestimated (overestimated) during the NW (SE) case. The size distribution of the
three ice hydrometeors assumed by all three sensitivity simulations (supplement Fig. S4)
reveals that the implementation of the BR mechanism produces elevated concentrations of





cloud ice crystals but at the same time increases the concentrations of snow particles. This is
the reason why the modeled total ice mass is also increased compared with the CNTRL
simulation.

Fig. 3c compares the simulated cloud LWC to the concurrent CDP observations at JFJ

station. The LWC values recorded during the NW case are highly variant, reaching up to 0.75
gm$^{-3}$, which is substantially higher than the respective maximum LWC observed during the SE
case (0.30 gm$^{-3}$). On 26 January before 12:00 UTC, all three sensitivity simulations predict
LWC > 1 gm$^{-3}$, which, however, cannot be validated against measurements due to missing
values in the CDP time series. Note that this period is excluded from the statistics derived in
Table 5. The CNTRL simulation is found to overestimate the cloud LWC, predicting 0.42
(0.25) gm$^{-3}$ in the third quartile, which is a factor of ~2 higher than the mean LWC observed
during the NW (SE) case (Table 5).

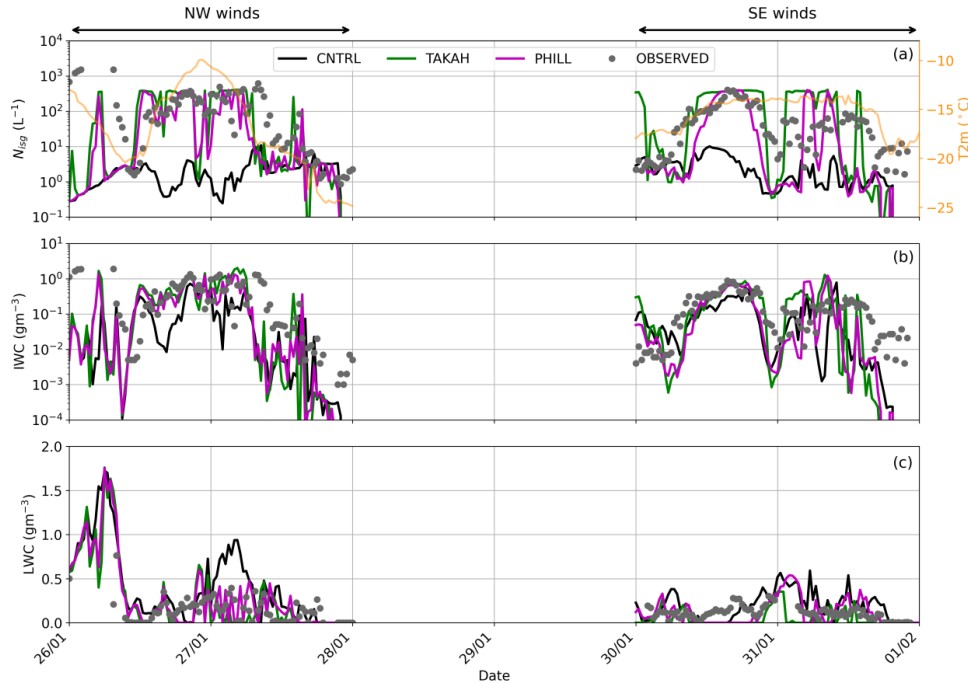

**Figure 3.** Time series of (a) total $N_{isg}$ and temperature at 2 m height (orange line), (b) IWC and
(c) LWC, predicted by the CNTRL (black line), TAKAH (green line) and PHILL (magenta
line) simulations between 26 January and 1 February 2014. The grey dots in all three panels
represent the 2D-S ICNCs, the inferred IWC and the CDP LWC measured at the JFJ station,
respectively. Note the logarithmic y-axes in panels a and b.





The modeled LWC in the 75th percentile is decreased by a factor of >2.5 (~1.5) in
TAKAH (PHILL) simulations (Table 5), improving the agreement with observations (Fig. 3c).
This reduction in LWC is expected, considering that the higher $N_{isg}$ produced when BR is
activated can readily deplete the surrounding droplets under liquid water subsaturated
conditions through the WBF process. This introduces a challenging environment to simulate,
as the model is sometimes seen to convert water to ice too rapidly, leading to cloud glaciation
(e.g., on 30 January after 12:00 UTC). Despite all sinks of cloud water (i.e., condensation
freezing, WBF or riming), observations at JFJ suggest that mixed-phase regions are generally
sustained (Lloyd et al., 2015). This is particularly true for the NW case, when the sufficiently
large updrafts caused by the steep ascent of the air masses help maintain the supersaturation
with respect to liquid water (Lohmann et al., 2016). PHILL can more efficiently sustain the
observed mixed-phase conditions compared to TAKAH, which frequently results in explosive
ice multiplication – especially during the SE case – leading to an underestimation of the LWC
(see Fig. 3c and Table 5).

**Table 5.** The 25th, 50th (median) and 75th percentiles of LWC (in gm$^{-3}$) time series.

| Simulation | 25th perc. | | Median | | 75th perc. | |
|---|---|---|---|---|---|---|
| | NW winds | SE winds | NW winds | SE winds | NW winds | SE winds |
| OBSERVED | $8.5\times10^{-3}$ | $70.0\times10^{-3}$ | 0.12 | 0.11 | 0.21 | 0.14 |
| CNTRL | $87.7\times10^{-3}$ | $26.0\times10^{-3}$ | 0.19 | 0.17 | 0.42 | 0.25 |
| TAKAH | $1.3\times10^{-10}$ | 0.0 | 0.01 | $6.7\times10^{-10}$ | 0.16 | 0.05 |
| PHILL | $6.3\times10^{-8}$ | 0.0 | 0.09 | 0.03 | 0.26 | 0.18 |
| BIPS10 | $82.0\times10^{-3}$ | $4.6\times10^{-3}$ | 0.18 | 0.12 | 0.33 | 0.24 |
| BIPS100 | $67.1\times10^{-3}$ | $13.1\times10^{-3}$ | 0.18 | 0.13 | 0.36 | 0.23 |
| BIPS100_PHILL | $6.3\times10^{-10}$ | 0.0 | 0.09 | 0.06 | 0.27 | 0.10 |


The time-averaged vertical profiles of number concentrations of cloud ice ($N_i$), graupel
($N_g$), snow ($N_s$) and total $N_{isg}$ are illustrated in Fig. 4 for the CNTRL, TAKAH and PHILL
simulations. $N_i$ (Fig. 4a) and $N_{isg}$ (Fig. 4d) are enhanced by more than 2 orders of magnitude in
TAKAH and PHILL compared to CNTRL. It is again obvious that TAKAH produces higher
concentrations than PHILL, at least in the lower 2 (1) km of the atmosphere during the NW
(SE) case. As discussed above, this regularly leads to overestimated $N_{isg}$ compared with the
observed amount of ice close to the surface. All three simulations, however, produce similar
results at heights above ~2.5 km, where the simulated temperatures drop well below –20 ºC





(supplement Fig. S5). This implies the greater importance of SIP through BR at heights below
2-3 km in the atmosphere (i.e., in the temperature range between ∼ –18 °C and ∼ –10 °C).
Graupel number concentrations (Fig. 4b) do not contribute much to the modeled ice
phase, especially during the SE case when the simulated $N_g$ is negligible compared with the $N_i$
and $N_s$ (Fig. 4c). In the M05 scheme, portion of the rimed cloud or rain water onto snow is
allowed to convert into graupel (Reisner et al. 1998), provided that snow, cloud liquid and rain
water mixing ratios exceed a threshold of 0.1, 0.5 and 0.1 g kg$^{-1}$, respectively. These mixing
ratio thresholds for graupel formation are arbitrary and might not be suitable for the examined
conditions, preventing the formation of graupel from rimed snowflakes (Morrison and
Grabowski, 2008). During the NW case, however, we can identify substantially higher $N_g$ than
the SE case, owing to the presence of sufficient supercooled liquid water especially during the
first half of 26 January. Activating the BR mechanism in TAKAH and PHILL generally
decreases the simulated $N_g$ in both cases (Fig. 4c), suggesting that break-up of graupel
contributes to ice multiplication.

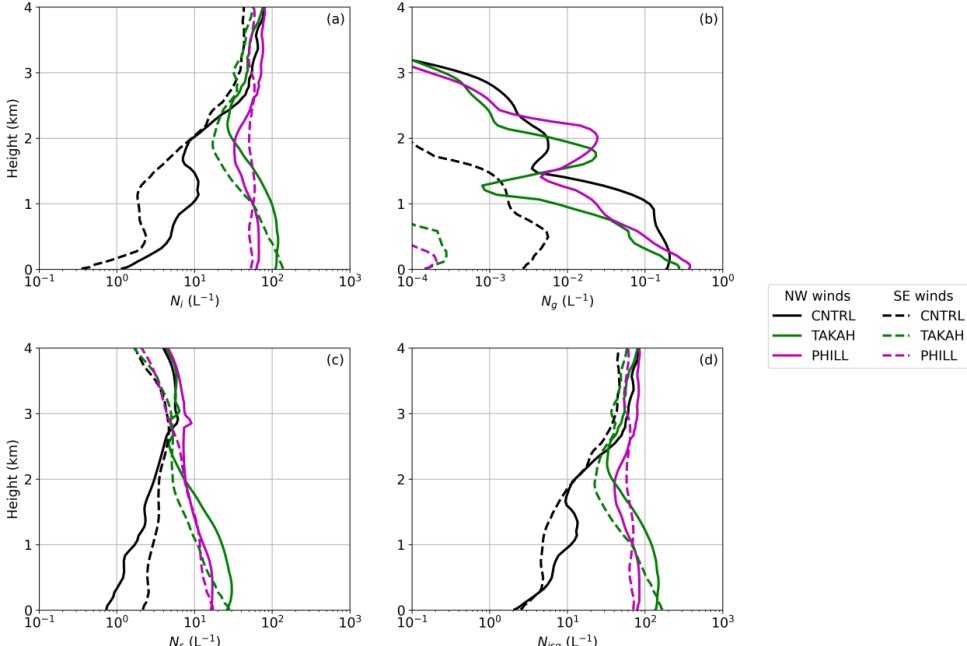


**Figure 4.** Mean vertical profiles of (a) $N_i$, (b) $N_g$, (c) $N_s$ and (d) total $N_{isg}$, predicted by the
CNTRL (black), TAKAH (green) and PHILL (magenta) simulations for the NW (solid lines)
and SE (dashed lines) cases. Note the different scale on the x axis of the $N_g$ vertical distribution.
The height is given in km a.g.l.

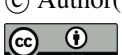




673    The mean vertical profile of $N_s$ (Fig. 4c) seems to follow the respective profile of $N_i$ (Fig.

674 4a). Unlike the graupel concentrations, including the BR mechanism is found to enhance $N_s$ up

675 to one order of magnitude compared to the CNTRL simulation. Focusing on a single model

676 time step when the BR mechanism is activated, the size distribution of snow particles shown

677 in the supplement Fig. S4 reveals that the increase in snow number concentrations can reach

678 up to 2 orders of magnitude during the NW case. This is a logical consequence of the increase

679 in number concentration of ice crystals, which are converting to snow particles after ice crystal

680 growth (i.e., cloud-ice-to-snow autoconversion), when surpassing a characteristic mean

681 diameter of 250 μm. This will be discussed in detail in the following section, which is focused

682 on PHILL simulation as it provides a slightly more accurate representation of the in-cloud

683 phase partitioning compared with TAKAH.

685 *3.1.1 Conditions favoring BR in the two considered events*

686 The temporal evolution of the vertical profiles of $N_{isg}$, IWC and LWC can provide valuable

687 insight on the drivers of enhanced ice formation in the wintertime alpine MPCs. Fig. 5 reveals

688 the presence of a seeder-feeder cloud system with sustained mixed-phase conditions confined

689 to levels below ~3 km (~1.5 km) in the NW (SE) case and a pure ice cloud aloft. Such

690 configurations are a well-known type of orographic multi-layer clouds that enhances

691 precipitation over mountains (e.g., Browning et al., 1974, 1975; Roe, 2005). Cloud

692 condensation is promoted by the synergy between a midlatitude frontal system and its

693 orographically induced ascent over the mountain range (Fig. 1). The separation between the

694 seeder and feeder clouds is often nonexistent, meaning that ice seeding can occur either in

695 layered clouds or internally within one cloud (Roe, 2005; Proske et al., 2021). In the first case,

696 which seems to occur here as well, there can be vertical continuum of cloud condensates

697 between the seeder and the feeder cloud due to precipitation of ice crystals from the higher-

698 level cloud (Fig. 5a). This means that the seeding ice crystals fall through subsaturated cloud-

699 free air before reaching the feeder region of the cloud and might sublimate. A remote-sensing

700 analysis to 11-year of data over Switzerland showed that in-cloud seeding occurs in 18% of the

701 observations, while the external seeder-feeder mechanism is present 15% of the time (Proske

702 et al., 2021) when the seeder is a cirrus cloud.

703    To illustrate the processes taking place during the two cases of interest, Fig. 6 displays

704 the tendency of primary and secondary ice production as well as the growth of ice particles





through deposition, riming and aggregation from the CNTRL and PHILL simulations at 17:00
(19:00) UTC on 26 (30) January. The vertical profiles on 26 January are taken within the
seeder-feeder event, while those on 30 January are taken when the high-level cloud associated
with the warm front has already passed the region of interest. Upon arrival of the frontal system
on 26 January, the CNTRL simulation indicates a rapid increase of the total $N_{isg}$ near cloud top
(Fig. 5a), which is not shown in the vertical profile of the primary ice production rates taken at
17:00 UTC (Fig. 6a). The ice particles consisting the seeder cloud are, therefore, formed far
from the JFJ station and seem to be advected over the domain of interest. Primary ice crystals
are formed in both cases below 2 km in the feeder cloud at temperatures lower than -30 ºC
through heterogeneous freezing (Fig. 6a). At these heights supercooled liquid water is also
present (Fig. 5c) and the newly formed ice particles start growing initially by vapor deposition
due to supersaturation with respect to ice, followed by riming (Fig. 6b). This is also indicated
by the increased IWC values closer to the ground (Fig. 5b).

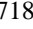

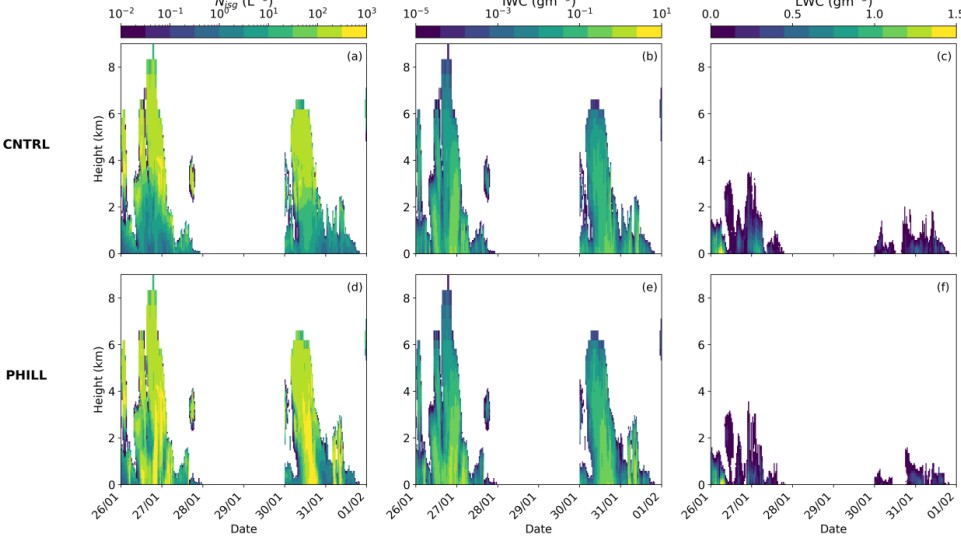

**Figure 5.** Time-height plots of total $N_{isg}$ (a, d), IWC (b, e) and LWC (c, f) produced by CNTRL
(top panel) and PHILL (bottom panel) simulations between 26 January and 1 February 2014.
The height is given in km a.g.l.

Focusing on the ice-seeding event of 26 January, the enhanced aggregation rate observed

at heights above ~2.5 km in the atmosphere indicates the enhanced collision efficiencies of the
precipitating ice particles while falling from the seeder cloud (Fig. 6c). Note that a portion of
the sedimented ice particles sublimates before reaching the feeder cloud at heights ~3-5 km,



indicating the prevailing unsaturated conditions in this layer (Fig. 6b). Within this layer the
aggregation of snowflakes weakens, while it is enhanced again when the falling hydrometeors
enter the feeder cloud. The bottom line is that, even under the simulated seeder-feeder events
the concentrations of ice particles reaching the ground in CNTRL simulation remain severely
underestimated (Section 3.1). Despite the low concentrations of ice crystals simulated by the
CNTRL simulation, the low-level cloud is glaciated more frequently during the SE than during
the NW winds case (Fig. 5c). This is probably because of the higher updraft velocities
prevailing until 28 January (Fig. 2d), preventing ice crystals from falling through the lower
parts of the cloud (Lohmann et al., 2016).

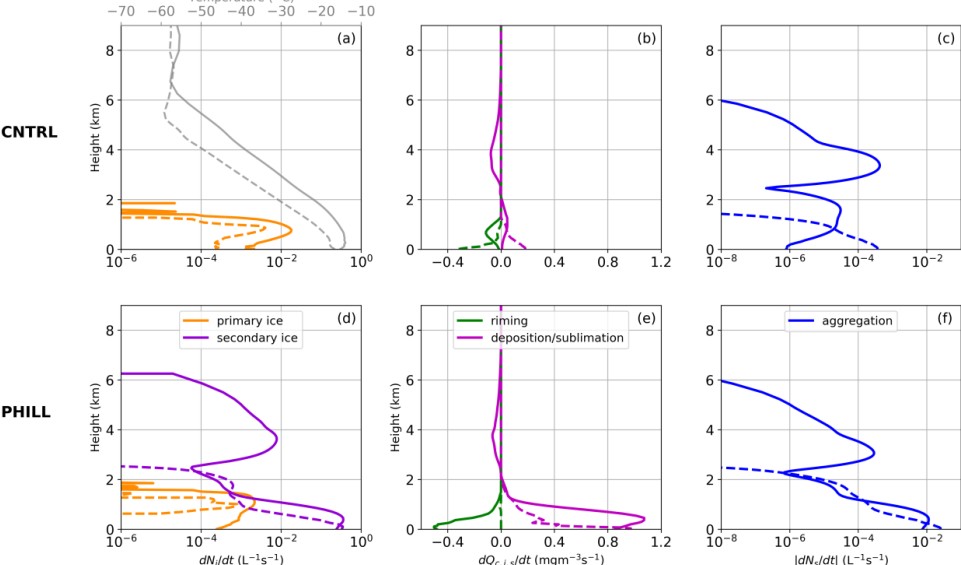


**Figure 6.** Vertical profiles of (a, d) primary and secondary ice production, (b, e) riming and
vapor deposition or sublimation and (c, f) snow aggregation produced by the CNTRL (top
panel) and PHILL (bottom panel) simulations at 17:00 UTC on 26 January (solid line) and at
19:00 UTC on 30 January (dashed line). The vertical profile of simulated temperature is also
superimposed in (a). The cloud liquid water content ($Q_c$) is shown in panels (b and e) to
represent the tendency due to riming, while the mass mixing ratio of the ice and snow species
($Q_i + Q_s$), are representing the relative tendencies due to vapor deposition or sublimation. Note
that the tendencies due to snow aggregation in (c, f) are presented in absolute values. The height
is given in km a.g.l.

Activating the BR mechanism along with the seeding of precipitating hydrometeors in

PHILL simulation shifts the simulated $N_{isg}$ towards higher concentrations that are found to
exceed 300 L$^{-1}$ in the lower-level part of the cloud (Fig. 5d). On 26 January the mode of the
cloud ice distribution shifts to slightly bigger sizes, while on 30 January the modal sizes





become almost an order of magnitude smaller compared with the CNTRL simulation (supplement Fig. S4). The enhanced concentrations of bigger ice particles simulated in the first case experience rapid growth through vapor deposition and riming (Fig. 6e) causing a slight increase in the simulated IWC (Fig. 5e) at the expense of the surrounding cloud droplets in the low-level feeder cloud (Fig. 5f). Nevertheless, the smaller ice particles simulated in the second case grow less efficiently through vapor deposition, while the explosive multiplication of ice through BR seems to fully glaciate the low-level cloud below ~1 km resulting in an almost zero riming rate (Fig. 6e). The reduced primary ice production rate observed during both case studies is a consequence of the depletion of liquid water when BR is considered (Fig. 6d).

The key difference between CNTRL and PHILL simulations is that the latter takes advantage of the enhanced ice particle growth through aggregation while falling to the feeder cloud below ~2 km, where large snowflakes coexist with smaller ice crystals (Fig. 4a, 6a, 6d). This allows for differential settling, which enhances collision efficiency facilitating ice multiplication through BR. This is the reason why the vertical profile of secondary ice formation agrees with the corresponding profile of aggregation during both case studies (Fig. 6d, 6f). On 26 January the first secondary ice particles start forming already within the seeder cloud with the contribution of SIP increasing considerably when reaching the feeder cloud, where the tendency due to SIP is more than 3 orders of magnitude higher than primary ice production (Fig. 6d). The significant role of SIP stands out also on 30 January at altitudes below 2 km. It is, therefore, essential to consider SIP though BR in the feeder cloud, in order to achieve the enhanced levels of ICNCs frequently observed within seeder-feeder events in the alpine region. This is in agreement with the observational study of Ramelli et al. (2021) on an ice-seeding case occurring in the region around Davos in the Swiss Alps. In this study, they proposed that SIP though HM and BR were necessary to explain the elevated ICNCs in feeder clouds.

A classification of the dominant type of precipitation was applied to the polarimetric data collected by a weather radar deployed at the Kleine Scheidegg station (2061 m a.s.l) during the SE case between 30 and 31 January (supplement Fig. S6). In the derived time series we can identify periods when individual ice crystals (not aggregated and not significantly rimed) dominate over the entire precipitation column followed by periods when a clear stratification is present with ice crystals aloft and mostly aggregates and rimed ice particles below. This stratification is observed on 30 January at 19:00 UTC when the model tendencies are extracted (dashed lines in Fig. 6). Allowing for the BR process in PHILL simulation results in a 2 orders of magnitude enhancement in the aggregation rates close to the ground, which can better





reproduce the signatures observed in the hydrometeor classification at that time. An increase
in the simulated aggregates and rimed particles is expected to increase orographic precipitation,
which is important given that these low-level feeder clouds are incapable of producing
significant amounts of precipitation. Indeed, the mean surface precipitation produced by
PHILL is 30% (10%) increased during the NW (SE) case compared with CNTRL (Fig. S7),
which is in contrast to Dedekind et al. (2021) where the activation of the BR process is found
to suppress the regions of strong surface precipitation. This was attributed to the limited
efficiency of the small secondary ice particles to grow sufficiently to precipitation sizes when
the local updrafts lift them to the upper parts of the cloud that were glaciated. The radar-based
hydrometeor classification reveals also the predominance of ice crystals at the beginning and
the end of the precipitating periods (e.g., on 30 January at 15:00-17:30 or 31 January at 04:30-
06:00), which is again more consistent with the vertical profile of $N_i$ produced by PHILL rather
than the CNTRL simulation (supplement Fig. S6, S8).

3.2 Sensitivity to the injection of ice crystals from the surface
In this section we examine if the surface-originating small ice particles could have the potential
to initiate and enhance ice particle growth in the near-surface MPCs present in our case studies.
Fig. 7 illustrates two additional WRF simulations – BIPS10 and BIPS100 – where the ice
crystal source applied to the first model level is equal to 10 and 100 $L^{-1}$, respectively (Table 2).
Note that these two sensitivity tests do not consider any SIP process to analyze the influence
of BIPS only. The total $N_{isg}$ values produced in BIPS10 are only slightly increased compared
to the CNTRL simulation and generally remain outside the observed range at JFJ (Fig. 7a). An
order of magnitude increase in the applied NBIPS is seen to enhance the modeled $N_{isg}$ during
both case studies, however our simulations are still lacking ice particles. This is particularly
evident during the NW winds case, where the simulated $N_{isg}$ varies most of the time around 10
$L^{-1}$, remaining an order of magnitude lower than the observations. During the SE case, the
model performance is slightly improved with the $N_{isg}$ reaching up to ~25 $L^{-1}$ in BIPS100, which
occasionally falls within the lower limit of the observed ICNC values (e.g., in the evening of
31 January). At times when the detected ICNCs remain quite low (i.e., on the order of 10 $L^{-1}$),
the contribution of blowing snow particles probably from the Aletsch Glacier is sufficient to
explain the observations at JFJ.
As indicated in Fig. 7b, during the NW flow the simulated LWC at the first model level
in BIPS10 and BIPS100 almost coincides with the CNTRL simulation of WRF. The three
sensitivity simulations are producing comparable median and quartile LWC values (Table 5),





with BIPS10 and BIPS100 producing median LWC values closer to the observed ones during
the SE flow. When comparing against the LWC values in the third quartile though, the two
simulations lead to an overestimation up to a factor of ~1.5 during both case studies. Given that
there is approximately a factor of >20 (5) difference between the modeled and observed ICNCs
during the NW (SE) winds case (Table 3), Fig. 7 overall reveals that the addition of a source
of ice crystals from the effect of blowing snow cannot account for the observed liquid-ice phase
partitioning in the simulated orographic MPCs.

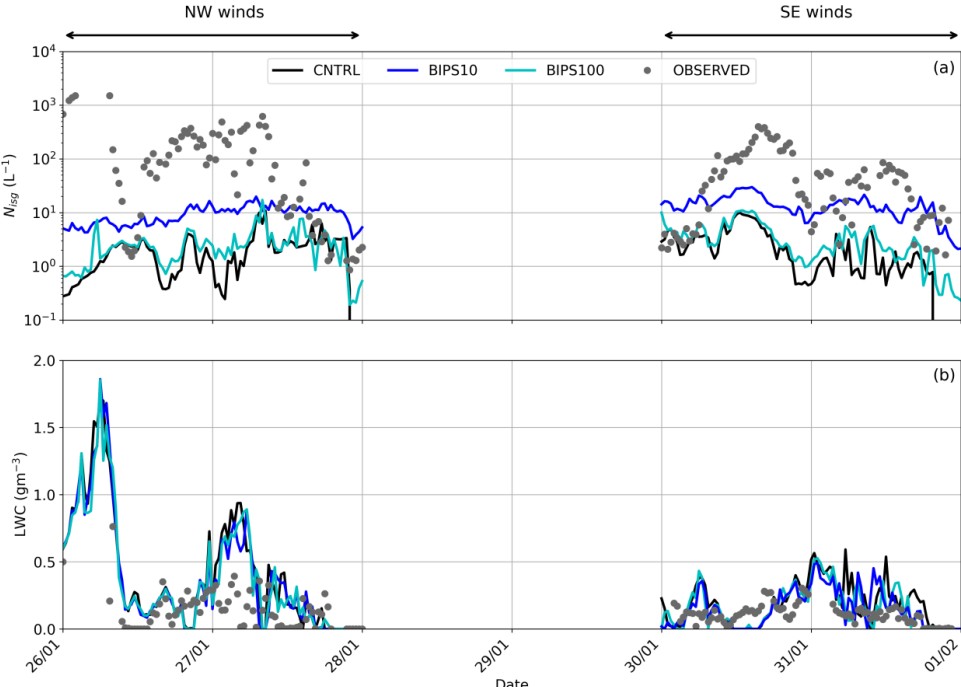

**Figure 7.** Time series of (a) total $N_{isg}$ and (b) LWC, predicted between 26 January and 1
February 2014 by the two sensitivity simulations accounting for the effect of blowing snow,
BIPS10 (cyan line) and BIPS100 (blue line).

Our findings are in contrast with the modeling study of Farrington et al. (2016), where a

different approach was proposed to include the surface effect on the ICNCs simulated with
WRF. In this study, a single model domain was used with a horizontal resolution of 1 km. To
account for the flux of hoar crystals being detached from the surface by mechanical fracturing,
Farrington et al. (2016) included a wind-dependent surface flux of frost flowers adapted from
Xu et al. (2013). Despite the improved performance of the WRF model in terms of predicted



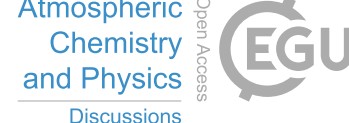

ICNCs and LWC, the wind-dependent formulation of the surface flux caused the modeled
ICNCs to become strongly correlated with the simulated horizontal wind speed – a behavior
that was not confirmed by the observations of Lloyd et al. (2015). Nonetheless, the highest
observed ICNCs at the beginning of the NW case correspond to the time when both the
observed and modeled wind speed is the strongest (Fig. 2c), implying that a wind-dependent
surface flux of BIPS could potentially elevate the simulated $N_{isg}$ to the observed levels at this
time.

3.3 The synergistic impact of BR and surface-induced ice crystals
It is deducible from the above discussion that the sole inclusion of a constant source of BIPS
in our simulations cannot efficiently bridge the gap between modeled and measured ICNCs.
Our aim in this section is to explore the combined effect of SIP through BR and blowing snow
on the simulated orographic MPCs, since both processes are deemed to be important when
trying to explain the high ICNCs observed in alpine environments. This is addressed in the
final sensitivity simulation, BIPS100_PHILL, the results of which are compared with the
CNTRL and PHILL simulations in Fig. 8.

In terms of the modeled ice particle concentrations, the combination of the simplified

blowing snow treatment and BR parameterization can account for most of the discrepancy
between modeled and observed ICNCs, particularly during the SE case (Fig. 8a), when the
simulation leads to best agreement with the observed interquartile values (Table 3).
BIPS100_PHILL and CNTRL generally differ by an average factor of ~100 (40) during the
NW (SE) case, with the former producing $N_{isg}$ values that are sometimes elevated by up to ~3
(2) orders of magnitude (Fig. 8a). Compared to PHILL setup, including a source of BIPS is
found to improve the modeled ICNCs close to the surface episodically – for instance in the
evening of 30 and 31 January, with the $N_{isg}$ in BIPS100_PHILL efficiently reaching the
observed levels (Fig. 8a). Note that BIPS can contribute to the modeled $N_{isg}$ even without the
presence of a near-surface orographic cloud (e.g., Geerts et al., 2015; Beck et al., 2018). For
instance, BIPS100_PHILL is the only sensitivity simulation producing high $N_{isg}$ values in the
evening of 27 and 31 January, when the low-level cloud is dissipated (Fig. 5c, f). In the former
case, however, the model results in an overestimate of the ICNCs, which is also observed
during the early hours of 30 January, suggesting that the applied source of ice crystals is
unrealistically high at this time.
As the mixed-phase conditions are sustained throughout both case studies (Fig. 8c), the
plume of ice crystals is mixed into an ice-supersaturated environment and, thus, BIPS are
expected to promote ice growth through their interaction with the surrounding supercooled
liquid droplets and (ice) supersaturated air. The number of BIPS reaching the cloud base might
not be large, but their presence is expected to further facilitate the action of the BR mechanism,
considering the depositional growth they will undergo within the supercooled boundary layer
cloud. This is illustrated for example with the concurrent increase in $N_{isg}$ and IWC observed on
30 January at approximately 21:00 UTC (Fig. 8a, 8b) in the presence of the low-level cloud
(Fig. 8c). Note that the elevated $N_{isg}$ caused by the addition of BIPS is not always followed by
an efficient increase in the simulated IWC. This can be observed for example on 27 January at
12:00 UTC or in the evening of 31 January (Fig. 8b).

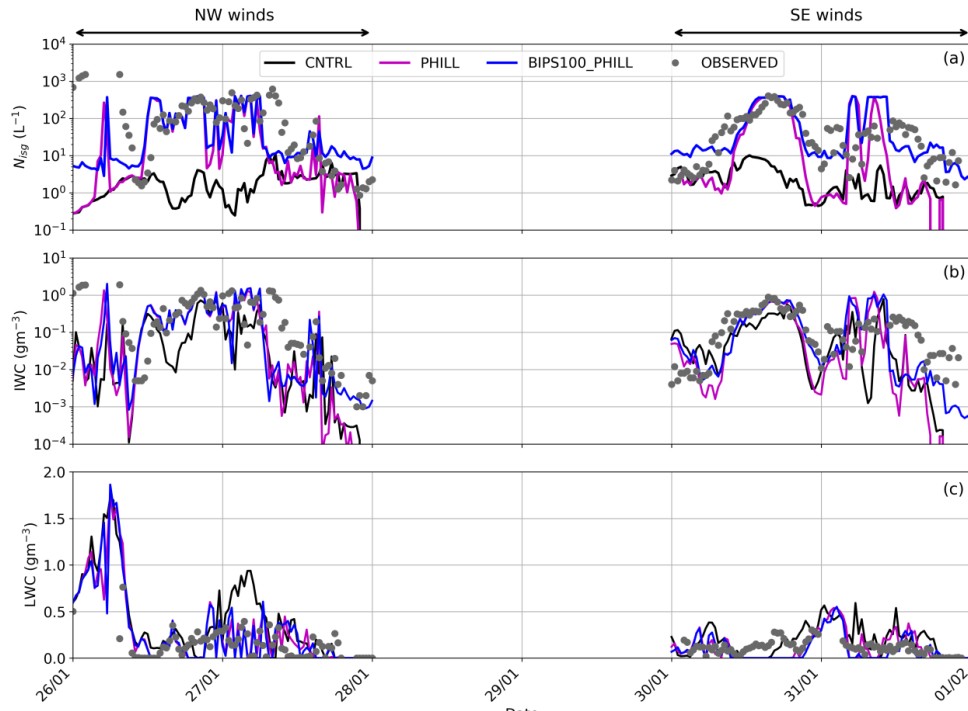

**Figure 8.** Time series of (a) total $N_{isg}$, (b) IWC and (c) LWC, predicted between 26 January
and 1 February 2014 by the sensitivity simulation BIPS100_PHILL (blue line), which
examines the combined effect of ice multiplication through BR and blowing snow.

A discrepancy between modeled and observed IWC was also highlighted in the study of
Farrington et al. (2016), and was attributed to the small sizes of the hoar frost particles assumed



(i.e., 10 μm). Although here BIPS are assumed to have sizes of 100 μm, still the
underestimation in the cloud IWC has not been overcome. This suggests that the applied source
of BIPS combined with the effect of SIP through BR shifts the ice particle spectra to smaller
sizes, which are not very efficient in riming and the WBF process and, thus, do not always
contribute to significant increases in IWC values. Overall, the interquartile values presented in
Table 4 reveal that BIPS100_PHILL and PHILL yield almost identical IWC values, suggesting
that the implementation of a constant source of BIPS does not further improve the
representation of the total ice mass despite the improvements in the simulated $N_{isg}$. Focusing
on the LWC values in the third quartile, though, including a source of BIPS results in better
agreement with the CLACE observations during the SE case, while it is shown to have little
effect on the cloud liquid phase during the NW case (Table 5). Despite the increase in the
modeled $N_{isg}$ observed in BIPS100_PHILL especially during the SE case, the liquid water in
the low-level orographic cloud is not further depleted (Fig. 8c). This is presumably because the
mean surface precipitation produced is also enhanced by almost ~20% compared to PHILL
(supplement Fig. S7), which seems to balance the excessive ice production.

One final point that is worth noting here is that there are still some certain periods when
BIPS100_PHILL fails to reproduce the observed range of ICNCs. This could imply the
potential contribution of additional ice multiplication processes to the observed ice particle
concentrations. Indeed, the seeder-feeder configuration observed in the examined case studies
could favor the fragmentation of sublimating hydrometeors while falling through an
subsaturated environment before entering the feeder cloud (e.g., Bacon et al., 1998). The so-
called "sublimational break-up" is an overlooked SIP process which is not yet described in the
M05 scheme. Also, note that the periods when the modeled ICNCs remain below the observed
ice number levels are mainly identified when the simulated temperature drops below –15 °C
and the wind speed exceeds 10 ms⁻¹ or even 20 ms⁻¹ (e.g., in the morning of 26 January or 27
January at around 12:00 UTC). This is when the incorporation of surface-based processes
becomes of primary importance. The simplified methodology we followed here although
instructive, yet it faces several limitations. For instance, the constant source of BIPS is
sometimes found to overestimate the modeled $N_{isg}$ and IWC. In order to accurately assess the
potential role of the snow-covered surfaces in elevating the simulated ICNCs, an improved
spatio-temporal description of the concentration and distribution of BIPS is required.
Furthermore, the applied ice crystal source is independent of some key parameters controlling
its resuspension, such as the horizontal wind speed, the updrafts or the friction velocity (e.g.,
Vionnet et al., 2013, 2014). For example, in the early morning hours of 26 January, the high





simulated horizontal and vertical velocities (Fig. 2c, 2d) are expected to loft significant BIPS
concentrations into the cloud layer, owing to enhanced mechanical mixing and momentum flux
close to the surface. Nonetheless, the contribution of the induced plume of BIPS remains
constant throughout the NW case study (Fig. 7a), which seems to lead to an underestimation
of the total ice particle concentration and mass. A more realistic parameterization of the BIPS
flux or the coupling with a detailed snowpack model would, therefore, be essential for a more
accurate representation of the effect of blowing snow.

## 4.    Summary and conclusions

This study employs the mesoscale model WRF to explore the potential impact of ice
multiplication processes on the liquid-ice phase partitioning in the orographic MPCs observed
during the CLACE 2014 campaign at the mountain-top site of JFJ in the Swiss Alps. The
orography surrounding JFJ channels the direction of the horizontal wind speed, giving us the
opportunity to analyze two frontal cases occurring under NW and SE conditions.
DS and BR mechanisms were implemented in the default M05 scheme in WRF, in
addition to the HM parameterization, which however remained inactive in the simulated
temperature range (–10 to –24 °C). The DS process is parameterized following either the latest
theoretical formulation developed by Phillips et al. (2018) or the more simplified
parameterization proposed by Sullivan et al. (2018a). Our sensitivity simulations revealed that
the DS mechanism is ineffective in the two considered alpine MPCs, even under the higher
updraft velocity conditions associated with the NW winds case study. This is due to a lack of
sufficiently big raindrops, necessary to initiate this process.
To parameterize the number of fragments generated per ice-ice collision we followed
again two different approaches: either the simplified temperature dependent formulation of
Takahashi et al. (1995) scaled for the size of the particle that undergo fragmentation
(Sotiropoulou et al., 2021a) or the more advanced physically-based Phillips et al. (2017a)
parameterization. Including a description of the BR mechanism is essential for reproducing the
ICNCs observed in the simulated orographic clouds, especially at temperatures higher than ~
–15 °C, where INPs are generally sparse. SIP through BR is found to enhance the modeled
ICNCs by up to 3 (2) orders of magnitude during the NW (SE) case, improving the model
agreement with observations. This ice enhancement can cause up to an order of magnitude
increase in the mean simulated IWC values compared with the CNTRL simulation, which is



attributed to the enhanced ice crystal growth and cloud-ice-to-snow autoconversion. The
increase in the simulated ICNCs also depletes the cloud LWC by at least a factor of 2 during
both cases, which is more consistent with the measured LWC values.
One of the most interesting outcomes of this study is the association of the enhanced BR
efficiency with the occurrence of in-cloud seeder-feeder events, which are commonly found in
Switzerland (Proske et al., 2021). While ice-seeding situations are associated with enhanced
orographic precipitation in the alpine region, the CNTRL simulation fails to reproduce the
elevated ICNCs reaching the ground. The falling ice hydrometeors experience efficient growth
through aggregation in the seeder part of the cloud, which is enhanced when reaching the feeder
cloud at altitudes below 2 km, where primary ice crystals form and grow through vapor
deposition and riming. Aggregation of snowflakes seems to be the major driver of secondary
ice formation in the examined seeder-feeder events. SIP though BR is initiated already within
the seeder cloud, while it becomes immensely important in the feeder cloud where its
production rate exceeds the one of primary ice formation. The increased generation of
secondary ice fragments does not always lead to ice explosion and cloud glaciation, as it is
followed by an enhancement in the precipitation sink owing to a shift in the ice particle
spectrum. Including a description of the BR mechanism is, therefore, crucial for explaining the
ice particle concentration and mass observed in the low-level feeder clouds.
To assess the potential role of blowing snow in the simulated orographic clouds, a
constant source of ice crystals was introduced in the first atmospheric level of WRF. Our results
indicate that blowing snow alone cannot explain the high ICNCs observed at JFJ, but when this
source is combined with the BR mechanism then the gap between modeled and measured
ICNCs is sufficiently bridged. The biggest influence of blowing snow is mainly detected at
times when the simulated temperatures are low enough ($< -15\ ^{\circ}$C), while the presence of a low-
level cloud is required for SIP to manifest. The concentrations of BIPS reaching the cloud base
are not high, but when they are mixed among supercooled liquid droplets they are expected to
grow, facilitating ice multiplication through BR. Nonetheless, including a wind-dependence or
a spatio-temporal variability in the applied ice crystal source would be essential to provide a
more precise description of the effect of blowing snow on the simulated clouds.
Overall, our findings indicate that outside the HM temperature range, a correct
representation of both secondary ice and an external ice seeding mechanism, which is primarily
precipitating ice particles formed aloft and to a lesser degree wind-blown ice from the surface,
will improve the accuracy of the liquid-ice partitioning in MPCs predicted by atmospheric





numerical models. More precisely, the implementation of SIP through BR can effectively shift
the number concentrations of ice particles in the right direction dictated by observations of
alpine MPCs, which is in turn critical not only for the determination of their optical properties
but also for the accurate estimate of precipitation patterns.

*Code and data availability.* The WRF outputs presented in this study will be made available at
https://zenodo.org/, while the updated Morrison scheme is available upon request. @Note by
authors: Data will be made available upon acceptance of final publication.

*Competing interests.* The authors declare no conflict of interest.

*Author Contributions.* PG and AN conceived and led this study with input from GS. EV helped
with the WRF configuration and setup. GS provided the updated microphysics scheme with
the detailed BR parameterizations. PG implemented the DS parameterizations with help from
GS, conducted the simulations, analyzed the results and, together with AN, wrote the main
paper. All authors contributed to the scientific interpretation and writing of the paper.

*Acknowledgements.* The authors would like to thank Gary Lloyd for providing the
microphysical measurements, as well as Jacopo Grazioli for collecting and pre-processing the
radar data. The authors are also thankful to Varun Sharma and Michael Lehning for the fruitful
discussions on the contribution of blowing snow in the alpine region.

*Financial support.* This research has been supported by the Horizon 2020 project FORCeS
(grant 821205) and the European Research Council project PyroTRACH (grant 726165). GS
received funding from the Swedish Research Council for Sustainable Development FORMAS
(project ID 2018-01760).





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
