# Peer review of "Secondary ice production processes in wintertime alpine 1 mixed-phase clouds 2"

_Atmospheric Chemistry and Physics, 2021_

## Referee Comment (RC1)

Review of "Secondary ice production processes in wintertime alpine mixed-phase clouds"

The paper uses data from CLACE-2014 and the WRF model to investigate possible explanations for the observations of high ice particle concentrations in mixed-phase orographic clouds. The findings suggest that surface sources of ice particles are not the main source, but the collision of ice particles with other ice particles, leading to fragmentation can explain the observed high ice particle concentrations. An enhancement of 3 orders of magnitude is seen in the simulations, which is broadly consistent with the observations.

I felt the introduction was too long. The material is good, but it felt more like a "literature synthesis" section, rather than an introduction. The introduction could be shorter, contextualising background information, stating the problem, and stating the response. The literature review is good, but I felt there was a bit too much detail for an introduction and it could be in a separate section after the introduction.

Page 10, equation 1: my understanding of the results of Takahashi et al. (1995) is that the ice fragments are produced during collisions with rime particles. Further, Vardiman (1978) also suggests it is more the rimed particles that produce fragments. Therefore, is it appropriate to say that collisions between cloud ice-snow and snow-snow produces fragments? In Takahashi et al.'s paper one of the ice spheres grew by collection of supercooled water drops. They say "our laboratory results support Takahashi (1993) hypothesis that ice crystals are generated by collision between large and small graupel". So I am unconvinced that collisions with un-rimed particles should produce fragments. I see the scaling in equation 2, but this is more to do with size, rather than degree of riming.

The second scheme (equation 3) appears to be more realistic, however it also requires more inputs, and it wasn't clear how these were calculated. For example, the rimed fraction of the most fragile ice particle – is this calculated in the model? How? Is it based on the observations (e.g. line 503)? This could be stated here for clarity. If it is based on the observations, how was it calculated? Or was it an estimate from images? If so, are the images statistically representative?

 Mode 2: "this mode has been studied only once in the laboratory study of Latham… " there is now another study, see James et al. (2021).

I think it would be worth mentioning where the Lawson DS parameterisation comes from. You mention the laboratory studies for other secondary ice mechanisms, but don't mention that Lawson et al. is based on fitting a curve in a model so that the model matches aircraft measurements. The shape of the curve is specified priori to have a D^4 dependence.

Reading through the rest of the results and discussion I am in agreement with the findings of the paper. Figure 3 is quite convincing that ice-ice collisional breakup should be important in these cases. Maybe the Takahashi parameerisation as implemented overestimates the number of ice particles compared to the Phillips implementation (because it is applied to all ice categories). I think this would be worth mentioning anyway, because it seems like applying it to all ice categories should lead to an overestimation.

I agree that the DS mechanism may not be strongly active in these cases because observations showed there were not many large drops present.

---

## Author Comment (AC1)

**Response to Referee #1**

The authors would like to thank the reviewer for carefully reading the revised manuscript and providing very thorough and constructive remarks to improve this paper. The replies to the comments are given below. Reviewer comments are highlighted in blue with our responses in black fonts.

I felt the introduction was too long. The material is good, but it felt more like a "literature synthesis" section, rather than an introduction. The introduction could be shorter, contextualising background information, stating the problem, and stating the response. The literature review is good, but I felt there was a bit too much detail for an introduction and it could be in a separate section after the introduction.
Thank you for this suggestion. We have now reduced the size of the introduction.

Page 10, equation 1: my understanding of the results of Takahashi et al. (1995) is that the ice fragments are produced during collisions with rime particles. Further, Vardiman (1978) also suggests it is more the rimed particles that produce fragments. Therefore, is it appropriate to say that collisions between cloud ice-snow and snow-snow produces fragments? In Takahashi et al.'s paper one of the ice spheres grew by collection of supercooled water drops. They say "our laboratory results support Takahashi (1993) hypothesis that ice crystals are generated by collision between large and small graupel". So I am unconvinced that collisions with un-rimed particles should produce fragments. I see the scaling in equation 2, but this is more to do with size, rather than degree of riming.
Thank you for raising and so thoughtfully and clearly articulating this excellent point. Applying the Takahashi parameterization to collisions between all ice categories may likely be the reason why the numbers of ice fragments are significantly overestimated. To evaluate this point, we included a new set of sensitivity simulations in the revised manuscript ("TAKAHrim") to contrast against the results of the TAKAH simulation. In TAKAHrim we first diagnose the presence of rime on the involved ice particles and then activate the Takahashi scheme only for collisions between rimed cloud ice/ snow and graupel particles. This new sensitivity simulation does not produce the large number of fragments seen in TAKAH, and in fact tends to follow the simulation using the Phillips parameterization. These new results and related insights are now included in the revised manuscript.

The second scheme (equation 3) appears to be more realistic, however it also requires more inputs, and it wasn't clear how these were calculated. For example, the rimed fraction of the most fragile ice particle – is this calculated in the model? How? Is it based on the observations (e.g. line 503)? This could be stated here for clarity. If it is based on the observations, how was it calculated? Or was it an estimate from images? If so, are the images statistically representative?
Parameters in the Phillips et al. (2017) parameterization that are not explicitly resolved in the Morrison microphysics scheme are the rimed fraction and the ice habit of colliding ice particles. The rimed fraction, is prescribed to a value of 0.4 (0.3) to account, respectively, for heavily and moderately rimed ice particles present under NW (SE) wind conditions. A high degree of riming

is expected in the simulated cases, as they both occur under ice-seeding situations, where large precipitating ice particles from the seeder clouds effectively gain mass in the mixed-phase zone through riming (see Figure 6 in the revised manuscript). Direct observations with balloon-borne measurements carried out within ice-seeding events in the region around Davos in the Swiss Alps support the presence of a large fraction of rimed particles and graupel (Ramelli et al., 2021). The higher riming degree is prescribed under NW-winds because the orographic forcing (i.e., vertical velocity) is stronger and helps maintaining mixed-phase conditions in the feeder clouds – which in turn promotes ice crystal growth through riming. That said, our results were not very sensitive to the rimed fraction.

The choice of ice habit is based on particle images collected during the CLACE 2014 campaign, showing the presence of non-dendritic sectored plates and oblate particles at temperatures ~ –15 ºC (Lloyd et al., 2015). Grazioli et al. (2015) also presented some examples of particle imagery produced by a two-dimensional stereoscopic (2D-S) shadow imaging probe, revealing the presence of heavily rimed hydrometeors, as well as highly oblate particles (probably columns or needles).

All the above points are elaborated in the text.

Mode 2: "this mode has been studied only once in the laboratory study of Latham… " there is now another study, see James et al. (2021).
Thank you for pointing this out. This sentence is now modified in the revised manuscript.

I think it would be worth mentioning where the Lawson DS parameterisation comes from. You mention the laboratory studies for other secondary ice mechanisms, but don't mention that Lawson et al. is based on fitting a curve in a model so that the model matches aircraft measurements. The shape of the curve is specified priori to have a D^4 dependence.
Good point, a more detailed description of the Lawson et al. (2015) parameterization is included in the revised text.

Reading through the rest of the results and discussion I am in agreement with the findings of the paper. Figure 3 is quite convincing that ice-ice collisional breakup should be important in these cases. Maybe the Takahashi parameterisation as implemented overestimates the number of ice particles compared to the Phillips implementation (because it is applied to all ice categories). I think this would be worth mentioning anyway, because it seems like applying it to all ice categories should lead to an overestimation.
Thank you once again for this comment. The issue about the Takahashi implementation is now raised in the revised manuscript.

I agree that the DS mechanism may not be strongly active in these cases because observations showed there were not many large drops present.
Indeed, 2D-S and CPI measurements did not show presence of drops large enough to initiate DS (Lloyd et al., 2015). WRF predictions are consistent with the observations, as the predicted raindrops do not exceed the 50 microns preventing the onset of the droplet shattering mechanism.

**References:**

Grazioli, J., Lloyd, G., Panziera, L., Hoyle, C. R., Connolly, P. J., Henneberger, J. and Berne, A.: Polarimetric radar and in situ observations of riming and snowfall microphysics during CLACE 2014, Atmos. Chem. Phys., 15, 13787–13802, doi:10.5194/acp-15-13787-2015, 2015.

Lawson, R. P., Woods, S. and Morrison, H.: The microphysics of ice and precipitation development in tropical cumulus clouds, J. Atmos. Sci., 72(6), 2429–2445, doi:10.1175/JAS-D-14-0274.1, 2015.

Lloyd, G., Choularton, T. W., Bower, K. N., Gallagher, M. W., Connolly, P. J., Flynn, M., Farrington, R., Crosier, J., Schlenczek, O., Fugal, J. and Henneberger, J.: The origins of ice crystals measured in mixed-phase clouds at the high-alpine site Jungfraujoch, Atmos. Chem. Phys., 15(22), 12953–12969, doi:10.5194/acp-15-12953-2015, 2015.

Phillips, V. T. J., Yano, J. I. and Khain, A.: Ice multiplication by breakup in ice-ice collisions. Part I: Theoretical formulation, J. Atmos. Sci., 74(6), 1705–1719, doi:10.1175/JAS-D-16-0224.1, 2017.

Ramelli, F., Henneberger, J., David, R., Bühl, J., Radenz, M., Seifert, P., Wieder, J., Lauber, A., Pasquier, J., Engelmann, R., Mignani, C., Hervo, M. and Lohmann, U.: Microphysical investigation of the seeder and feeder region of an Alpine mixed-phase cloud, Atmos. Chem. Phys., 21, 6681–6706, doi:10.5194/acp-21-6681-2021, 2021.

Sotiropoulou, G., Ickes, L., Nenes, A. and Ekman, A.: Ice multiplication from ice–ice collisions in the high Arctic: sensitivity to ice habit, rimed fraction, ice type and uncertainties in the numerical description of the process, Atmos. Chem. Phys., 21, 9741–9760, doi:10.5194/acp-21-9741-2021, 2021.

---

## Author Comment (AC2)

**Response to Referee #2**

The authors would like to thank the reviewer for the feedback. The replies to the comments are given below. Reviewer comments are highlighted in blue with our responses in black fonts.

1. Line 211: Please clarify if the whole 2D-S spectrum was considered for estimating the observed ice number concentrations. It is good practice to remove particles smaller than 200 um while reporting the number concentration of ice particles from the 2DS probe as the shattering removal algorithm may not remove all the artifacts. Authors can add a few lines on these uncertainties.

   Thank you for bringing up this point. One of the main outcomes of our study is the fact that secondary ice production shifts the ice particle size distributions to smaller sizes, which can then alter the whole microphysical evolution of the simulated mixed-phase clouds. When the effect of the collisional break-up process is considered in our simulations, the mode of the cloud ice size spectrum drops below 100 um (supplement Fig. S4), which would be completely unconstrained from observations if ice particle sizes less than 200 um would be discarded.

   We of course understand the concern nevertheless. The 2D-S data were compared with the rotator position and the inlet velocity – if the rotator was not pointing into the wind or the inlet was blocked with rime the data would have been filtered. Analysis of probe imagery carried out by the experimentalists during the CLACE 2014 campaign and inter-arrival time (IAT) histograms did not reveal the presence of shattered particles, likely because of the much lower velocity at which this probe was aspirated ($\sim 15$ ms$^{-1}$) compared to those during aircraft deployments. We will make this point more clear in the text.

2. Line 258: Define ERA5

   Thank you, this is now corrected in the text.

3. Line 261: Define MODIS

   Thank you, corrected.